# Closed-loop Long-horizon Robotic Planning via Equilibrium Sequence Modeling

**Jinghan Li** [1]   **Zhicheng Sun** [1]   **Yadong Mu** [1]

## Abstract

In the endeavor to make autonomous robots take actions, task planning is a major challenge that requires translating high-level task descriptions to long-horizon action sequences. Despite recent advances in language model agents, they remain prone to planning errors and limited in their ability to plan ahead. To address these limitations in robotic planning, we advocate a self-refining scheme that iteratively refines a draft plan until an equilibrium is reached. Remarkably, this process can be optimized end-to-end from an analytical perspective without the need to curate additional verifiers or reward models, allowing us to train self-refining planners in a simple supervised learning fashion. Meanwhile, a nested equilibrium sequence modeling procedure is devised for efficient closed-loop planning that incorporates useful feedback from the environment (or an internal world model). Our method is evaluated on the VirtualHome-Env benchmark, showing advanced performance with improved scaling w.r.t. inference-time computation. Code is available at `https://github.com/Singularity0104/equilibrium-planner`.

## 1. Introduction

Recent advances in large language models (LLMs) have spurred tremendous progress in robotic planning (Huang et al., 2022; Li et al., 2022; Singh et al., 2023; Driess et al., 2023; Ahn et al., 2023; Huang et al., 2023; Zhao et al., 2023; Hu et al., 2024). Based on their extensive world knowledge, LLM agents seem close to autonomously performing robotic tasks, such as in household scenarios. However, growing evidence shows that existing LLM agents struggle with task planning (Kaelbling & Lozano-Pérez, 2011) that decom-

poses a high-level task into mid-level actions. While this problem requires long-horizon planning as well as consideration of environmental feedback, LLMs are often limited by: (1) *unidirectional dependency*: due to autoregressive generation, previous tokens cannot attend to future tokens, resulting in limited ability to plan ahead (Wu et al., 2024); (2) *lack of error correction* for existing outputs, unless with a heavy system 2; (3) *fixed forward process* hindering the allocation of more inference computation to further improve planning performance. These inherent limitations of LLMs inhibit closed-loop long-horizon robotic planning.

To address the above challenges of LLM planners in closed-loop long-horizon planning, we advocate the approach of self-refinement (Welleck et al., 2023; Shinn et al., 2023; Kim et al., 2023b; Madaan et al., 2023) that iteratively improves a previously generated plan. The reasons behind are threefold: (1) *bidirectional dependency*: since the output is conditioned on a previous draft plan, it can attend to all tokens in the plan (from an old version), thus improving its ability to plan ahead; (2) *internal error correction* which allows implicit self-correction in a forward pass without an explicit, heavy system 2; (3) *dynamic computation allocation* by iterating through a self-refinement process until convergence. However, such a self-refining strategy imposes significant training difficulties because it requires backpropagation through infinite self-refining steps (Werbos, 1990). This may be seen as an extreme case that reflects some of the general challenges in teaching LLMs to plan and reason. Existing solutions include curating process supervision (Uesato et al., 2022; Lightman et al., 2024) or applying reinforcement learning (Zelikman et al., 2024; Jaech et al., 2024; Kumar et al., 2025; Guo et al., 2025), but they are considerably more complex than supervised training and remain unproven in the absence of efficient verifiers.

This work proposes a simple supervised learning framework for planning via self-refinement. Specifically, we formulate the self-refining process as a fixed-point problem that recursively refines the plan until the equilibrium point, as illustrated in Figure 1. While the forward process of this fixed-point problem could be solved efficiently using root-finding methods, more interestingly, its backpropagation can be *skipped* since its gradient is explicated by the implicit function theorem (Krantz & Parks, 2002) as in deep equilibrium models (Bai et al., 2019; Geng & Kolter, 2023).

[1]Peking Unviersity, China. Correspondence to: Yadong Mu <myd@pku.edu.cn>.

*Proceedings of the 42nd International Conference on Machine Learning*, Vancouver, Canada. PMLR 267, 2025. Copyright 2025 by the author(s).

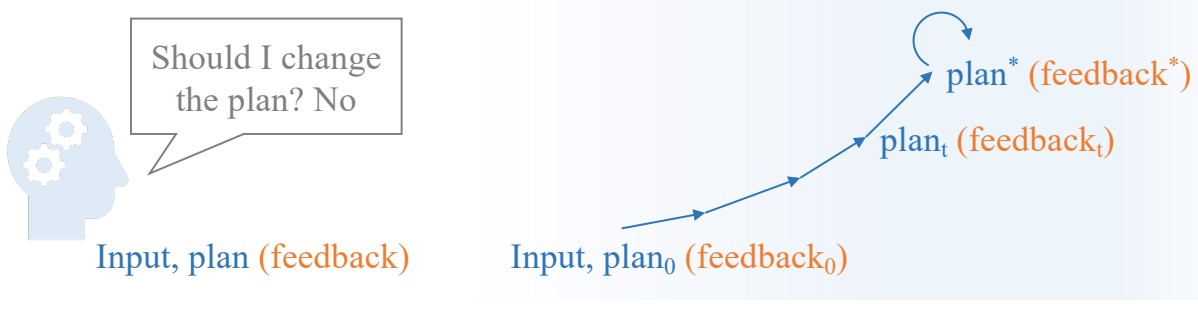

(a) Equilibrium in planning

(b) Iterative planning until an equilibrium is reached

Figure 1: Illustration of the equilibrium point in planning. We view planning as a self-refinement process in which the ideal plan emerges as an equilibrium point, remaining unchanged by any refinement attempts even with newer information (*e.g.* feedback from the environment or a world model). This enables us to tackle robotic planning from an optimization perspective around its equilibrium, bypassing the need for sophisticated reinforcement learning.

It is noted that the derived gradient term may be further simplified through a Jacobian-free approximation (Fung et al., 2022) to facilitate training. These analytical techniques allow end-to-end supervised training of the LLM planner to accomplish self-refinement without the need for additional verifiers or reward models in reinforcement learning-based counterparts, greatly enhancing simplicity and practicality. And after training, our equilibrium model-based planner dynamically allocates more inference computation based on the number of iterations needed to solve the equilibrium, thereby achieving better planning performance.

Another important cue for self-refinement in robotic tasks is closed-loop feedback from the environment. To efficiently incorporate environmental feedback, we devise a nested equilibrium sequence modeling procedure consisting of inner and outer loops, where the inner loop iteratively refines a plan using previous feedback, while the outer loop updates the feedback by interacting with the environment. This enables closed-loop planning from even a few environmental interactions. Moreover, the nested equilibrium solving process is accelerated by reusing the previously derived equilibrium. We further implement the above design within an LLM agent framework, seamlessly integrating the equilibrium model-based planner, an experience memory buffer containing past plans and feedback, and a world model to estimate feedback in the absence of environmental interactions, thus allowing the planning system to operate effectively in closed-loop long-horizon robot task planning scenarios. The core contributions of our work are as follows:

- We present equilibrium sequence modeling, a simple training approach for self-refining LLMs based on equilibrium models, allowing for end-to-end supervised learning without additional verifiers or reward models.

- A nested equilibrium solving process is proposed to ef-

ficiently incorporate closed-loop feedback into the equilibrium sequence modeling, reusing previous equilibrium solutions to alleviate inference computation. It is further implemented with a world model to improve practicality.

- Our method is evaluated on the VirtualHome-Env benchmark (Puig et al., 2018; Liao et al., 2019), demonstrating its advantageous performance with better scaling w.r.t. inference computation than tree-based alternatives.

## 2. Related Work

**LLMs for Planning.** LLMs demonstrate outstanding capabilities in robotic planning (Silver et al., 2024; Zhang et al., 2023; Nayak et al., 2024; Wang et al., 2024). Scaling up inference computation to improve LLMs' performance on planning and reasoning tasks has received increasing attention (Brown et al., 2024; Snell et al., 2025; Wu et al., 2025; Jaech et al., 2024; Guo et al., 2025). Precedent techniques involving chain-of-thought (Wei et al., 2022; Zelikman et al., 2022; 2024), repeated sampling (Wang et al., 2023; Brown et al., 2024) and tree search (Yao et al., 2023a; Zhao et al., 2023) showed preliminary results with handcraft system 2. Alternatively, a method called self-refinement (Welleck et al., 2023; Shinn et al., 2023; Kim et al., 2023b; Madaan et al., 2023) suggests recursively refining the existing LLM output in an autonomous manner, but it relies heavily on prompting or sophisticated reinforcement learning. To fully exploit its potential, we propose an end-to-end optimization method for self-refinement via deep equilibrium models.

**Deep Equilibrium Models** (Bai et al., 2019) are infinite-depth neural networks specified by fixed-point problems $x^* = f_\theta(x^*)$, where $f_\theta$ is an equilibrium layer. While their inference can take infinite steps by the fixed-point iteration, their gradients are estimated using implicit differentiation (Krantz & Parks, 2002) without backpropagating

through all layers, thus enabling memory-efficient training. They have been extensively applied to tasks such as visual understanding (Bai et al., 2020; 2022) and image generation (Pokle et al., 2022; Geng et al., 2023; Bai & Melas-Kyriazi, 2024). In this paper, we apply the fixed-point formulation of deep equilibrium models to the self-refinement process in LLM planners, allowing for simple supervised training to refine themselves. More detailed introduction to deep equilibrium models is presented in Appendix A.

# 3. Method

We study the problem of robot task planning that aims to decompose a high-level task description into long-horizon mid-level action sequences (Kaelbling & Lozano-Pérez, 2011). In the following, we first introduce the closed-loop robotic planning problems (Section 3.1). Then we discuss the limitations of LLM planners in self-refinement (Section 3.2) and address them with a novel equilibrium sequence modeling scheme (Section 3.3). This framework is adapted to closed-loop feedback with efficient designs in Section 3.4. Finally, practical implementations are presented in Section 3.5.

## 3.1. Problem statement

We assume that the agent runs in a fully observed environment with coarse feedback. Given a high-level task described in natural language instruction $I_h$ and the corresponding environment information $Env$, the agent is required to decompose $I_h$ into a sequence of low-level actions $\{a_1, a_2...a_n\}$ as an action plan to accomplish its subgoals $\{g_1, g_2, ...g_m\}$, where $a_n$ are semantic actions from the given action space and $g_m$ are goal-oriented conditions that are invisible to the agent. In terms of closed-loop task planning, we allow the agent to interact with the environment many times and receive feedback for adjustment.

## 3.2. Preliminaries on Self-Refinement

The prevailing LLMs are intrinsically limited in planning, as their unidirectional dependency results in limited capability to plan ahead (Wu et al., 2024), and the lack of error correction hinders closed-loop planning. These reasons call for alternative mechanisms to address robot task planning.

Recently, Welleck et al. (2023); Shinn et al. (2023); Kim et al. (2023b); Madaan et al. (2023) proposed self-refinement, which uses an LLM $f_\theta$ to iteratively refine the previous LLM output. This strategy naturally addresses the above limitations, since it introduces bidirectional token dependency and a dynamic error correction mechanism. Formally, let $x_t$ denote a draft plan and $c_t$ denote context (e.g. environmental feedback), then planning may be viewed as a self-refinement process as follows:

$$x_{t+1} = f_\theta(x_t, c_t). \tag{1}$$

However, self-refinement via prompting (Shinn et al., 2023; Kim et al., 2023b; Madaan et al., 2023) has been found to be very limited by Huang et al. (2024). Alternative training-based methods require careful curation of training sequences (Welleck et al., 2023; Havrilla et al., 2024) or reinforcement learning (Qu et al., 2024; Kumar et al., 2025) and are therefore difficult to implement, even for domains with efficient verifiers such as coding and math. Overall, they remain deficient for robotic planning compared to system 2-based alternatives, as shown in Hu et al. (2024).

## 3.3. Self-Refinement as An Equilibrium Model

To address the training inefficiency of self-refinement approaches, this section proposes equilibrium sequence modeling, a simple supervised training scheme for teaching LLM planners to self-refine through the lens of deep equilibrium models (Bai et al., 2019; Geng & Kolter, 2023).

Let us first consider a simplified scenario of self-refinement without environmental feedback, namely that the context $c_t$ is fixed, e.g. to a predefined system message $c$. Then, the self-refinement process in Equation (1) reduces to a fixed-point problem concerning only the plan $x_t$. Denote the initial plan by $x_0 = \varnothing$ and the equilibrium plan, i.e. the endpoint, by $x^*$, then its trajectory can be expressed as:

$$(x_0, c) \rightarrow \ldots \rightarrow (x_t, c) \rightarrow \ldots \rightarrow (x^*, c). \tag{2}$$

Although its forward process is tractable with existing root-solving techniques, such as the classic fixed-point iteration or alternative numerical methods (Broyden, 1965; Anderson, 1965), its training requires recurrent backpropagation through multiple self-refining steps (Werbos, 1990). This results in an extremely inefficient and unstable computational process where end-to-end training fails.

Instead, we approach it directly from an analytical perspective. Assuming access only to outcome supervision $L(\cdot, y)$ on the plan, e.g. its distance to the ground truth plan $y$, then self-refinement is formulated as an optimization problem minimizing the loss function of the equilibrium plan:

$$\begin{aligned} \min_\theta \quad & L(x^*, y) \\ \text{s.t.} \quad & x^* = f_\theta(x^*, c). \end{aligned} \tag{3}$$

Interestingly, the above optimization problem can be solved without backpropagating over the entire inference process. As the following theorem indicates, we can directly differentiate through its fixed point regardless of the solution path, with only a constant computational and memory cost.

**Theorem 3.1.** (*Implicit Function Theorem (Bai et al., 2019; Krantz & Parks, 2002*)) *Assuming that $\left(I - \frac{\partial f_\theta}{\partial x^*}\right)$ is invertible, then the loss gradient of Equation (3) w.r.t. $\theta$ is given by:*

$$\frac{\partial L}{\partial \theta} = \frac{\partial L}{\partial x^*} \left(I - \frac{\partial f_\theta}{\partial x^*}\right)^{-1} \frac{\partial f_\theta}{\partial \theta}. \tag{4}$$

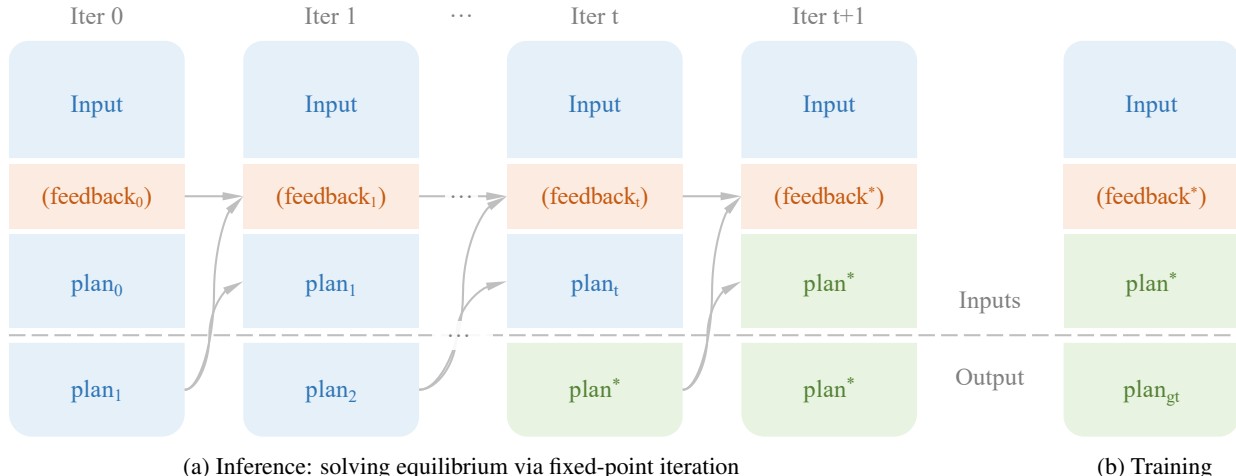

(a) Inference: solving equilibrium via fixed-point iteration  (b) Training

Figure 2: Illustration of equilibrium sequence modeling with two alternating steps: (a) Prior to training, the model undergoes iterative inference to reach an equilibrium plan. (b) Then, it is pushed away from the equilibrium towards the ground truth by a supervised loss. This teaches the model to self-refine by mapping a suboptimal equilibrium plan to a better plan.

Its proof is given in Appendix A.4. It is noteworthy that the inverse Jacobian term $A = (I - \frac{\partial f_\theta}{\partial x^*})^{-1}$ within the above gradient is difficult to compute exactly, for which existing work often approximates through the damped fixed-point unrolling or the Neumann series (Geng et al., 2021b). For computational efficiency, we drop the inverse Jacobian term using $A \approx I$ as in Fung et al. (2022); Geng et al. (2021a); Choe et al. (2023), the latter work having been validated on Transformer-based LLMs:

$$\frac{\partial L}{\partial \theta} = \frac{\partial L}{\partial x^*} \frac{\partial f_\theta}{\partial \theta}. \qquad (5)$$

For a comprehensive introduction to equilibrium models and this approximation, please refer to Appendix A.

**Equilibrium Sequence Modeling.** Based on the simplified gradient estimation, we reformulate its training into a supervised learning problem. According to the chain rule, the derived gradient is exactly the gradient of the following optimization problem associated with the equilibrium $x^*$:

$$\min_\theta \quad L(f_\theta(x^*, c), y). \qquad (6)$$

This new formula represents a new equilibrium sequence modeling scheme that can be implemented in two alternating steps: (1) In the first step, we solve the fixed-point problem by iterative inference of LLM based on the previous output tokens $x_t$ until the new output tokens converges, yielding an equilibrium plan $x^*$. (2) In the second step, the equilibrium $x^*$ is paired with the ground truth plan $y$ as a training sequence, which is used as in the standard supervised finetuning pipeline to teach the LLM to self-refine. The two-step procedure is illustrated in Figure 2.

It features two intuitive advantages: (1) instead of directly regressing the ground truth, it gently adjusts the equilibrium

point, which reduces overfitting compared to the vanilla supervised finetuning; (2) by guiding the LLM to map a suboptimal plan $x^*$ to a better plan $y$, it teaches self-refinement via a simple supervised loss, without requiring additional value functions or reward models (Welleck et al., 2023; Havrilla et al., 2024; Qu et al., 2024; Kumar et al., 2025).

### 3.4. Equilibrium Models with Feedback

This section extends the derived equilibrium sequence modeling to a more practical scenario where the environment may provide some closed-loop feedback, *e.g.* failure details, during plan execution. Such auxiliary information would be an effective cue for planners to further refine their plan.

To take into account environmental feedback, we consider an adaptive context $c_t$ that is influenced by the plan $x_t$ rather than fixed. Then, the previously considered equilibrium solving process of Equation (2) should be revised as an iterative process coupling the plan $x_t$ with the feedback $c_t$, starting from $x_0 = c_0 = \varnothing$:

$$(x_0, c_0) \rightarrow \ldots \rightarrow (x_t, c_t) \rightarrow \ldots \rightarrow (x^*, c^*). \qquad (7)$$

After the modification, the existing derivations only hold when we neglect the derivatives related to $c^*$. Fortunately, this is a natural choice due to the non-differentiability of most feedbacks. Therefore, the equilibrium planner can be trained in a similar supervised way as in Equation (6) and Figure 2, and after training it would be able to self-refine based on the latest feedback just by forward passes. However, iteratively interacting with the environment to obtain feedback is costly and may not be recoverable. In response, we devise a nested equilibrium solving scheme for more efficient closed-loop planning.

---

**Algorithm 1** Inference of Equilibrium Planner

---

**Require:** planner $f_\theta$, environment or world model Env, number of iterations $N$.
  Initialize a start point $x_0$ and feedback $c_0$.
  **for** $i \leftarrow 0$ to $N$ or converged **do**
    Solve inner equilibrium loop to obtain $x_t^*$. $\qquad\qquad\qquad\qquad\qquad\qquad\qquad\qquad\qquad\qquad$ ▷ Equation (8)
    Update next plan $x_{t+1}$ and feedback $c_{t+1}$ with Env. $\qquad\qquad\qquad\qquad\qquad\qquad$ ▷ Equations (9) and (10)
**Ensure:** generated plan $x^*$.

---

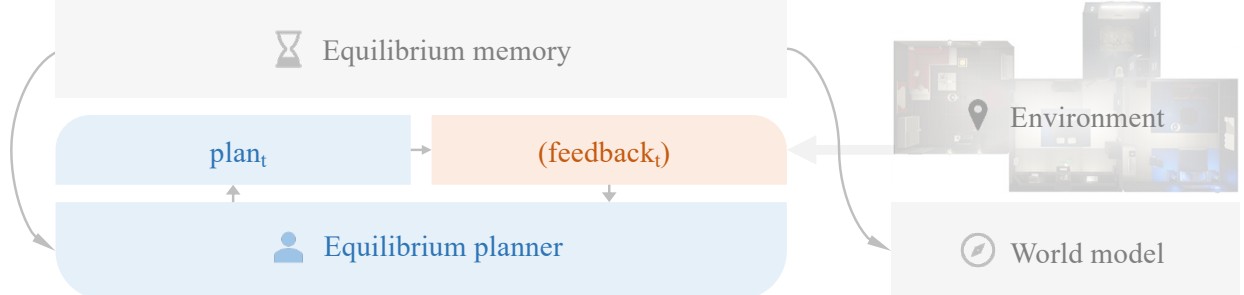

Figure 3: Illustration of our proposed framework. It incorporates (1) a memory containing all equilibrium experiences during inference, (2) a self-refining planner trained on equilibrium plans alongside the ground truth, and (3) a world model trained on experiences to simulate environmental feedback. Three modules work synergistically for closed-loop planning. The planner interacts with the environment to output plan$_t$ with equilibrium sequence modeling and the equilibrium plans with feedback are stored in the equilibrium memory. During training, The planner is trained on these equilibrium plans together with the ground truth while the world model is trained on the past experiences stored in the equilibrium memory.

**Nested Equilibrium Solving.** Inspired by the introspection process in human daily life, we propose to divide equilibrium solving into a nested loop process. The inner loop *introspects* on the existing plan and feedback and takes no action, while the outer loop interacts with the environment to update the feedback. Formally, each inner loop is an equilibrium solving process with fixed feedback $c_t$:

$$\begin{cases} x_t^1 = f_\theta(x_t^0, c_t) \\ \dots \\ x_t^* = f_\theta(x_t^*, c_t). \end{cases} \qquad (8)$$

Thanks to this inner-loop introspection mechanism, our equilibrium model can be more efficient in closed-loop planning, outperforming with fewer environmental interactions.

**Reusing Equilibrium Solution.** Another efficiency bottleneck is the equilibrium solving. Considering that its speed depends largely on the initial plan, it is unnecessary to restart from $\varnothing$ every time. Therefore, we accelerate equilibrium solving by reusing the previously derived equilibrium plan as the starting point of the next iteration, similar to Bai et al. (2022); Bai & Melas-Kyriazi (2024):

$$x_{t+1} = x_t^*. \qquad (9)$$

which corresponds to the starting point $x_{t+1}^0$ of the inner loop. Similarly, history feedback could be reused across

different inner loops. This is achieved by initializing the context of the next inner loop by concatenating the previous feedback and the latest feedback (paired with its plan):

$$c_{t+1} = (x_{t+1}, \text{Env}(x_{t+1})) \parallel c_t, \qquad (10)$$

where $\parallel$ denotes concatenation. The nested inference procedure with reuse of equilibrium is described in Algorithm 1.

### 3.5. Practical Implementation

This section discusses the implementation of the proposed equilibrium planner. To enable effective training while interacting with the environment, the following two modules are carefully devised to complement the planner: an experience memory that caches all equilibrium plans and their feedback during equilibrium solving, and a world model to estimate the feedback in the absence of environmental interactions. The complete planning framework is illustrated in Figure 3 and the specific implementation details are explained below.

**Equilibrium Experience Memory.** During the training process, our equilibrium model interacts with the environment only when the inner loop reaches the equilibrium point. This results in a small number of equilibrium points, which may not be sufficient for supervised training. To improve our training efficiency and stability, we opt to cache all previously obtained equilibrium points, along with their

Table 1: Performance on VirtualHome-Env without correction. Our planner achieves state-of-the-art performance in most evaluations. Note that the Exec. metrics are marked in gray because they are already high and can easily exceed 99% with simple automated rules (by truncating illegal output). See Table 8 in the appendix for full comparisons with Tree-Planner.

| | Novel Scene and Task | | | Novel Scene | | | Novel Task | | |
|---|---|---|---|---|---|---|---|---|---|
| | Exec. | SR | GCR | Exec. | SR | GCR | Exec. | SR | GCR |
| *GPT-3.5 API:* | | | | | | | | | |
| Zero-shot Planner | 16.49 | 1.07 | 1.52 | - | - | - | - | - | - |
| ProgPrompt | 35.04 | 12.54 | 19.99 | - | - | - | - | - | - |
| Iterative-Planner | 44.54 | 27.04 | 33.25 | - | - | - | - | - | - |
| Tree-Planner$_{N=25}$ | 55.74 | 28.33 | 39.96 | - | - | - | - | - | - |
| Tree-Planner$_{N=50}$ | 49.01 | 28.14 | 35.84 | - | - | - | - | - | - |
| *Finetuned Llama 3 8B:* | | | | | | | | | |
| Supervised | 93.55 | 24.19 | 32.55 | 96.84 | 41.05 | 49.81 | 95.94 | 26.07 | 35.53 |
| Tree-Planner$_{N=25}$ | 95.16 | 38.71 | 63.18 | 96.08 | 51.58 | 69.45 | 95.50 | 40.38 | **63.75** |
| Tree-Planner$_{N=50}$ | 94.94 | 38.71 | 63.50 | 96.06 | 51.58 | 69.54 | 95.40 | 39.74 | 63.29 |
| Ours | 90.32 | **40.32** | **65.40** | 95.79 | **65.26** | **79.47** | 93.38 | **41.88** | 62.76 |

Table 2: Performance on VirtualHome-Env with up to 10 corrections. Our planner consistently leads in SR and GCR performance. In particular, the comparison with SELF-REFINE confirms the effectiveness of our new training method.

| | Novel Scene and Task | | | Novel Scene | | | Novel Task | | |
|---|---|---|---|---|---|---|---|---|---|
| | Exec. | SR | GCR | Exec. | SR | GCR | Exec. | SR | GCR |
| *GPT-3.5 API:* | | | | | | | | | |
| Local Replan | 79.66 | 37.46 | 51.90 | - | - | - | - | - | - |
| Global Replan | 82.09 | 37.93 | 52.46 | - | - | - | - | - | - |
| Tree-Planner$_{N=25}$ | 89.13 | 35.30 | 56.65 | - | - | - | - | - | - |
| Tree-Planner$_{N=50}$ | 88.26 | 41.58 | 59.55 | - | - | - | - | - | - |
| *Finetuned Llama 3 8B:* | | | | | | | | | |
| SELF-REFINE | 96.77 | 43.55 | 65.18 | 92.63 | 54.74 | 70.24 | 94.44 | 39.96 | 62.37 |
| Tree-Planner$_{N=25}$ | 95.16 | 41.94 | 56.49 | 96.08 | 55.79 | 68.82 | 95.50 | 42.09 | 57.83 |
| Tree-Planner$_{N=50}$ | 94.94 | 43.55 | 58.91 | 96.06 | 58.95 | 70.00 | 95.40 | 43.38 | 59.79 |
| Ours | 91.94 | **56.45** | **76.63** | 97.89 | **77.89** | **87.07** | 92.31 | **54.91** | **74.18** |

environmental feedback, in an experience memory. Thereafter, these equilibrium points can be sampled repeatedly for versatile training purposes. For example, for the planner, we randomly sample a batch of equilibrium points at each training epoch, which are paired with the ground truth for supervised training. In particular, the most recent equilibrium points are sampled more frequently to reduce distribution shift. Next, we describe another crucial component.

**Internal Feedback from World Model.** Due to inefficiency of interacting with the environment, we construct a world model (Ha & Schmidhuber, 2018) to provide the necessary feedback in closed-loop planning. Our world model takes the environmental context, task instruction and current plan as inputs and predicts some basic types of feedback. This definition is slightly simpler than the commonly used world model, which requires simulation of the environmental states, and therefore may be easier to train. Concretely,

we implement the world model with an LLM and finetune it on the planner's equilibrium feedback over all iterations for better generalizability. And during inference, we alternate between using real and generated feedback at each iteration for a good balance between performance and efficiency.

## 4. Experiments

### 4.1. Experimental Settings

**Benchmark.** The VirtualHome-Env benhmark (Puig et al., 2018; Liao et al., 2019) is adopted during the experiments. It contains 1360 long-horizon tasks with ground truth action sequence annotations (average length 10.8) and provides updated scene graphs after each action, allowing simulation of closed-loop feedback. To analyze the generalizability, we divided the dataset into a training set and three test subsets, including the novel scene set, the novel task set, and the

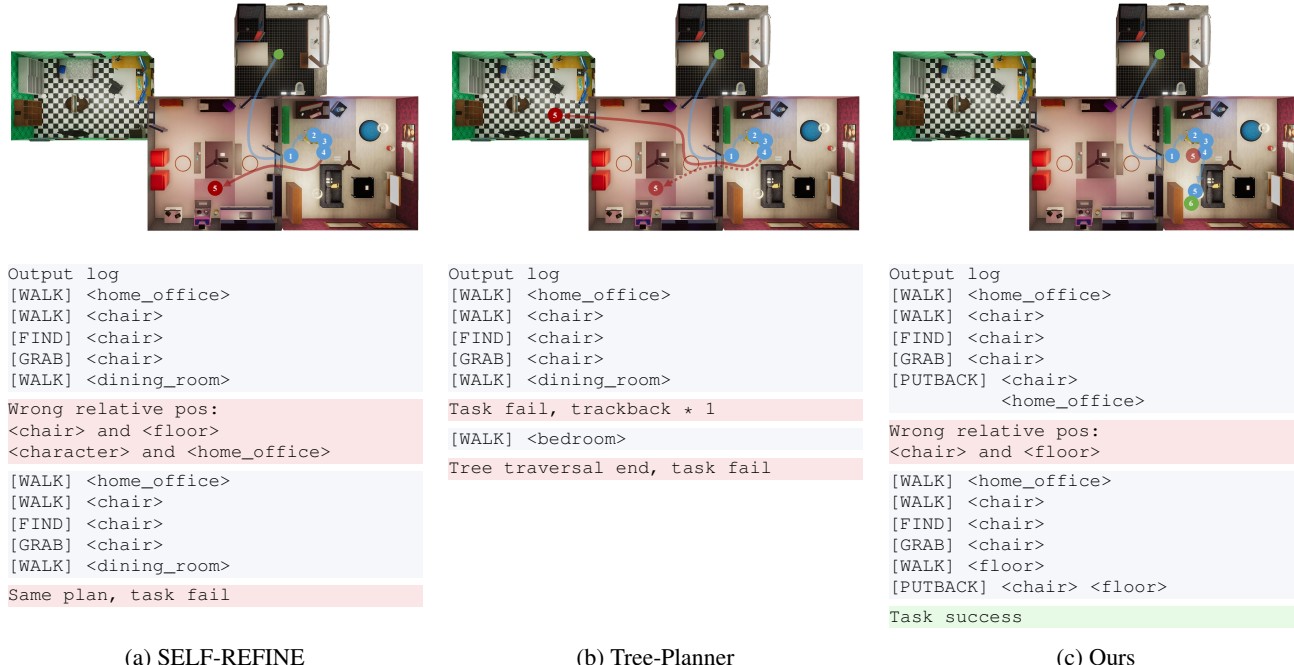

Figure 4: Visualization of our self-correction process compared to the baselines SELF-REFINE and Tree-Planner. The task instruction is "Take a comfortable chair and place it in the entrance hall".

novel scene and task set. More statistics and examples about VirtualHome-Env can be found in Appendix B.1.

**Metrics.** We use executability (Exec.), success rate (SR), goal conditions recall (GCR) following Hu et al. (2024). Exec. evaluates whether the plan can be executed in given the environment, SR refers to whether the goal is accomplished, and GCR measures the proportion of goal conditions achieved. To examine closed-loop planning capabilities, we study two test settings, without error correction or with up to 10 corrections, the latter allowing self-correction based on environmental feedback. We also evaluate computational efficiency by measuring TFLOPS at inference.

**Baselines.** Our method is compared with Tree-Planner (Hu et al., 2024), SELF-REFINE (Madaan et al., 2023), and a supervised finetuned planner. They are all reproduced using finetuned Llama 3 8B (Dubey et al., 2024) (the original Llama cannot achieve meaningful SR due to format errors). In addition, we consider several baseline methods that call the GPT-3.5 API, including ProgPrompt (Singh et al., 2023), Zero-shot Planner (Huang et al., 2022) and two self-refining planners, Local Replan (Raman et al., 2022; Guo et al., 2023) and Global Replan (Shinn et al., 2023). Their results are for reference only. See Appendix B.2 for details.

**Implementation Details.** Our implementation is consistent with the baseline methods by finetuning from Llama 3 8B (Dubey et al., 2024) on the VirtualHome-Env training

set (paired with the equilibrium points). The number of finetuning epochs is set to 6, and the learning rate is 0.0002. The world model is finetuned on all planner interactions for 5 epochs using the same learning rate. A greedy LLM sampling strategy is used in later refinement steps until convergence. Moreover, we implement the KV cache to speed up inference. Further details are provided in Appendix B.3.

### 4.2. Main Results

The experimental results in the two planning setups, without correction or with up to 10 corrections, are summarized in Tables 1 and 2. Overall, our method achieves the leading performance on the majority of metrics. Specifically, the experimental results show that: (1) Even without error correction, our self-refining process still brings a significant improvement of 14% on SR in the novel scene subset, with other metrics superior or comparable to the previous leading method. (2) By incorporating environmental feedback, our approach improves all metrics by more than 11% and up to 19%, showing clear advantages. (3) Similar to the existing finetuning-based methods, our generated plans exhibit a high executability of over 90%, which can be improved to 99% by simply truncating illegal overlength outputs. These results clearly confirm the advantages of our approach.

In particular, Table 2 compares the efficacy of our training scheme, equilibrium sequence modeling, against alternative self-correction methods. It shows more than 11% improve-

Table 3: Effectiveness of different types of feedback. They are measured under the constraint of up to 10 rounds of internal or external feedback. Our trained planner is able to take into account various types of feedback to refine the plan.

| World model | Env. | Novel Scene and Task | | | Novel Scene | | | Novel Task | | |
| --- | --- | --- | --- | --- | --- | --- | --- | --- | --- | --- |
| | | Exec. | SR | GCR | Exec. | SR | GCR | Exec. | SR | GCR |
| | | 88.71 | 33.87 | 59.98 | 96.79 | 49.47 | 66.60 | 93.80 | 34.62 | 59.06 |
| | ✓ | 83.87 | 51.61 | 75.13 | 96.84 | 75.79 | 85.79 | 92.31 | **56.62** | **75.53** |
| ✓ | | 90.32 | 40.32 | 65.40 | 95.79 | 65.26 | 79.47 | 93.38 | 41.88 | 62.76 |
| ✓ | ✓ | 91.94 | **56.45** | **76.63** | 97.89 | **77.89** | **87.07** | 92.31 | 54.91 | 74.18 |

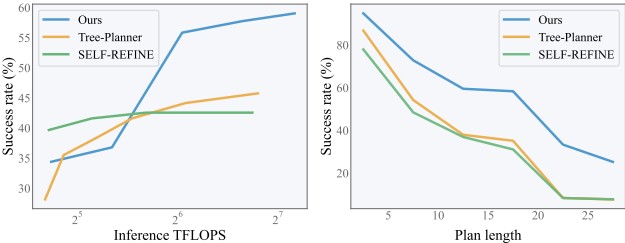

(a) Scaling of performance w.r.t. inference-time computation

(b) Comparison of performance over different planning horizons

Figure 5: Performance analysis vs. inference computation and plan length. Our method shows leading scaling w.r.t. inference computation and long-horizon planning capabilities. The inference computation is measured by TFLOPS, and we consider KV cache when computing inference TFLOPS.

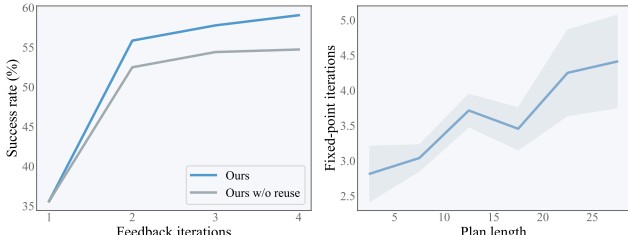

(a) Improving efficiency through reusing equilibrium solutions

(b) Dynamic compute allocation for more complex plans

Figure 6: Ablation study on computational efficiency. Our method reuses equilibrium solutions and dynamically allocates inference compute to improve performance-efficiency tradeoff. The latter is measured by the number of fixed-point iterations before convergence with mean and std.

ment over the prompting-based self-refinement method SELF-REFINE (Madaan et al., 2023) and more than 12% improvement over the system 2-based Tree-Planner (Hu et al., 2024). The former stems from the effectiveness of our training objective, while the latter is due to the flexible self-correction mechanism without extensive manual design. Meanwhile, we maintain a simplistic supervised finetuning fashion similar to the compared methods, without intricate reinforcement learning. These validates the effectiveness of equilibrium sequence modeling in robot task planning.

### 4.3. Visualization

Figure 4 further compares our self-correction process with the baseline methods. As can be seen, our method is better at incorporating environmental feedback to improve the plan, while the baselines fail by simply repeating the previous plan, or by making only local adjustments that are insufficient. The rationale behind is that our method can flexibly take into account feedback through forward passes of LLMs, allowing arbitrary changes based on its knowledge. In contrast, the tree-based alternative requires backtracking in a tree, which is costly and does not fully exploit the verbalized knowledge in the feedback. More qualitative results are illustrated in Figures 10 to 14 of the appendix.

### 4.4. Ablation Study

This section validates the effectiveness of our method in performance and efficiency. See Appendix C for more results.

**Effectiveness of various feedback.** As can be observed in Table 3, incorporating external feedback from the environment or internal feedback from the world model consistently improves performance. Even though the world model does not provide as much improvement as the real environment, it also increases performance by over 3%. In particular, the synergy of both types of feedback yields the highest performance on most of the metrics, further confirming their effectiveness. In the following analysis, we will focus on our method using only environmental feedback for simplicity.

**Scaling of performance.** Here, we follow Brown et al. (2024); Snell et al. (2025); Wu et al. (2025); Jaech et al. (2024); Guo et al. (2025) in considering the scaling w.r.t. inference computation. The results in Figure 5a show that our method achieves better performance-computation tradeoff along with leading scaling w.r.t. inference computation. Thus, more inference budget can be allocated to improve its performance. Furthermore, in Figure 5b, we show that its performance advantage is largely due to better long-horizon planning capabilities, achieving more than twice the success rate of baselines on extremely long plans (length>20).

Table 4: Convergence analysis. They summarize the number of iterations for LLMs to reach a fixed point over 10 runs on 60 random tasks with the equilibrium solving prompts. Each of them can reach convergence in a few steps.

| | Mean | Std | Min | Max |
|---|---|---|---|---|
| Original Llama-3-8B | 6.70 | 2.12 | 3.22 | 14.98 |
| Original Qwen-2.5-0.5B | 4.80 | 2.13 | 2.69 | 9.43 |
| Original Qwen-2.5-7B | 7.68 | 5.03 | 2.46 | 16.78 |
| Supervised Finetuned Llama | **2.46** | **0.88** | **2.02** | **3.80** |
| Ours | 3.02 | 1.61 | 2.32 | 4.07 |

Table 5: Robustness to disturbed environment. During inference, random feedback is injected into the environmental feedback according to the noise ratio. Our model remains stable, demonstrating robustness to noisy feedback.

| | Both Novel | | Novel Scene | | Novel Task | |
|---|---|---|---|---|---|---|
| Noise ratio | SR | GCR | SR | GCR | SR | GCR |
| 0% | 51.61 | **75.13** | **75.79** | 85.79 | **56.62** | **75.53** |
| 5% | 51.61 | 74.30 | **75.79** | 85.40 | 54.27 | 73.89 |
| 10% | **53.23** | 73.43 | 74.74 | **85.84** | 53.85 | 73.09 |
| 20% | 50.00 | 73.10 | 73.68 | 83.22 | 54.49 | 71.80 |

**Computational efficiency.** Although our planner training is slower than baselines (36h vs. 12h) due to the equilibrium solving process for synthesizing training pairs ($\approx$24h), it exhibits a competitive inference efficiency. For example, our method takes 16h to evaluate, while Tree-Planner takes 24h. This can be attributed to our design of reusing equilibrium in nested equilibrium solving, As illustrates in Figure 6a, it accelerates the convergence, achieving better performance ($>$55%) with significantly fewer interactions. Furthermore, Figure 6b shows that our planner dynamically allocates compute for tasks of different complexity.

**Fixed-point convergence.** Table 4 shows the number of iterations for an LLM to reach its fixed point, aggregated over 10 initial plan for each of 60 random tasks. All LLMs considered, including the original Llama and Qwen, the supervised finetuned Llama, and our model, can converge to a fixed point within a few iterations, regardless of the initial sequence. This is partly due to our greedy sampling strategy (described in Appendix B.3) that reduces the LLMs' randomness, after which they tend to repeat themselves (please see Figures 4a, 10a and 11a) and converge easily.

**Robustness to noisy feedback.** Table 5 shows the robustness of our model when the environment is disturbed by noise. The case where the model only receives environmental feedback is considered. During inference, we randomly replace some of the feedback with incorrect feedback, such as incorrect feedback types or incorrect corresponding objects and actions. As shown in the results, our model exhibits stable performance under small amounts of noise ($\leq$10%), demonstrating its robustness in a disturbed environment.

## 5. Conclusion

This work proposes an equilibrium model-based LLM planner that is capable of self-refining plans from external and internal feedback. Unlike existing self-refinement methods based on prompting or sophisticated reinforcement learning, our proposed equilibrium sequence modeling allows simple supervised training of the self-refining planners. Moreover, it also enables the planner to efficiently incorporate environ-

mental feedback or a world model for closed-loop planning. We implement the proposed approach on the VirtualHome-Env benchmark, and the experimental results suggest that it can dynamically allocate inference-time computation to achieve state-of-the-art robot task planning performance.

## Impact Statement

This paper presents a supervised learning framework for improving the closed-loop long-horizon capabilities of LLM agents, with the potential to complement the prevailing reinforcement learning-based or prompting-based frameworks. However, we note that any such improvement in planning capabilities should be treated with caution. For example, the LLM agent could autonomously create sub-goals that are threatening to humans, *e.g.*, gaining more control. This calls for further research into LLM interpretability and safety.

**Acknowledgements.** The work is supported by an internal grant of Peking University (2024JK28), a grant from China Tower Corporation Limited. We thank Xingjian Bai and Liyuan Wang for their helpful discussions.

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

The appendices are organized as follows. First, the background of deep equilibrium models is discussed in Appendix A. Then, we describe benchmarks in Appendix B.1, baselines in Appendix B.2, and our implementation details in Appendix B.3. Lastly, additional experimental results and limitation analysis are provided in Appendices C and D.

## A. Background

### A.1. Deep Equilibrium Models

Traditional neural networks are constructed by explicitly stacking layers $f^{(i)}$, which can be limited in their expressiveness due to the fixed number of layers and predetermined forward process. Instead, *implicit models* are defined by an underlying dynamic system to be solved, such as an ordinary differential equation (Chen et al., 2018), a controlled differential equation (Kidger et al., 2020), a stochastic differential equation (Kidger et al., 2021) or a fixed-point problem (Bai et al., 2019).

*Deep equilibrium models*, first introduced in Bai et al. (2019), are a representative class of implicit models characterized by fixed-point problems. Given an input $c$ and a function $f_\theta(x, c)$ such as a Transformer block (Bai et al., 2019) or a Transformer (Geng et al., 2023), deep equilibrium models define infinite-level stacking of this function $x_{i+1} = f_\theta(x_i, c)$ with $i = 0, 1, \ldots, L$ and $L \to \infty$ by solving the solution $x^*$ to the following fixed-point equation defined by $f_\theta$ and $c$:

$$x^* = f_\theta(x^*, c) \tag{11}$$

The forward pass of deep equilibrium models is root solving for the fixed-point problem. A common choice is the *fixed-point iteration* method, which starts from an initial guess $x_0$ and iteratively applies the transformation $x_{t+1} = f_\theta(x_t, c)$ until convergence. A sufficient condition for its convergence is if $f_\theta$ is a contraction mapping w.r.t. $x$, namely its Lipschitz constant is less than one (Banach, 1922), which could be relaxed by the well-posedness condition in El Ghaoui et al. (2021). More advanced root solvers include Broyden's method (Broyden, 1965) or Anderson acceleration (Anderson, 1965).

### A.2. Training Deep Equilibrium Models

Unlike traditional neural networks, whose gradient requires backpropagation through time (Werbos, 1990) at high memory and computational cost, the gradient of deep equilibrium models is computed analytically without differentiating over its forward pass. Given an equilibrium point $x^* = f_\theta(x^*, c)$ and a loss function $L(x^*, y)$, the loss gradient w.r.t. the model parameters $\theta$ is provided by the implicit function theorem (Krantz & Parks, 2002; Bai et al., 2019) as follows:

$$\frac{\partial L}{\partial \theta} = \frac{\partial L}{\partial x^*} \left( I - \frac{\partial f_\theta}{\partial x^*} \right)^{-1} \frac{\partial f_\theta}{\partial \theta}. \tag{12}$$

Its proof is given in Appendix A.4. Due to the challenge of exactly computing the inverse Jacobian term $A = (I - \frac{\partial f_\theta}{\partial x^*})^{-1}$ in the above gradient, existing work often approximate it via the damped fixed-point unrolling or the Neumann series (Geng et al., 2021b). Recently, Fung et al. (2022); Geng et al. (2021a) propose to approximate the inverse Jacobian term by $A \approx I$, the former proving it under strong theoretical assumptions. In practice, dropping the inverse Jacobian/Hessian has been used extensively in *one-step gradient* (Bolte et al., 2023; Luketina et al., 2016; Finn et al., 2017; Liu et al., 2019; Garima et al., 2020) and shown to be effective on Transformer-based LLMs (Choe et al., 2023).

### A.3. Transformer-based Deep Equilibrium Models

Deep equilibrium models are initially proposed on Transformer architecture (Vaswani et al., 2017) for language modeling tasks (Bai et al., 2019). This seminal work considers a Transformer block as the basic unit $f_\theta$ in the equilibrium model. Then, Geng et al. (2021a) investigates improvements over the Transformer block by replacing self-attention with matrix decomposition. They also introduce one-step gradient based on the approximation of $A \approx I$ for efficiency and stability, assuming that the Lipschitz condition apply to a large number of matrix decomposition methods.

Recently, following the prevalence of Diffusion Transformers (Peebles & Xie, 2023), deep equilibrium models are extended to image generation tasks. Geng et al. (2023) propose generative equilibrium Transformers consisting of two modules, one using Transformer as the basic unit $f_\theta$ in the equilibrium model. Their method yields advanced one-step image generation results. Bai & Melas-Kyriazi (2024) replace most of the intermediate Transformer blocks with an equilibrium model, thus significantly reducing the number of parameters and memory usage for training and inference.

## A.4. Proof of Implicit Function Theorem

**Theorem A.1.** *(Implicit Function Theorem (Bai et al., 2019; Krantz & Parks, 2002)) Let $L : \mathbb{R}^n \times \mathbb{R}^n \to \mathbb{R}$ be a differentiable loss function, and let $f_\theta : \mathbb{R}^n \times \mathbb{R}^p \to \mathbb{R}^n$ be a differentiable function parameterized by $\theta \in \mathbb{R}^q$. Consider the following optimization problem:*

$$
\begin{aligned}
\min_\theta \quad & L(x^*, y) \\
s.t. \quad & x^* = f_\theta(x^*, c).
\end{aligned}
\tag{13}
$$

*where $x^*, y \in \mathbb{R}^n$, and $c \in \mathbb{R}^p$. If $\left(I - \frac{\partial f_\theta}{\partial x^*}\right)$ is invertible, then the loss gradient w.r.t. $\theta$ is given by:*

$$
\frac{\partial L}{\partial \theta} = \frac{\partial L}{\partial x^*} \left(I - \frac{\partial f_\theta}{\partial x^*}\right)^{-1} \frac{\partial f_\theta}{\partial \theta}.
\tag{14}
$$

*Proof of Theorem 3.1.* To derive the loss gradient w.r.t. $\theta$, we begin by differentiating the equilibrium condition $x^* = f_\theta(x^*, c)$ with respect to $\theta$. Applying the chain rule, we have:

$$
\frac{\partial x^*}{\partial \theta} = \frac{\partial f}{\partial \theta} + \frac{\partial f}{\partial x^*} \frac{\partial x^*}{\partial \theta}.
\tag{15}
$$

Given that $\left(I - \frac{\partial f_\theta}{\partial x^*}\right)$ is invertible, we can rearrange the above equation and solve for $\frac{\partial x^*}{\partial \theta}$:

$$
\frac{\partial x^*}{\partial \theta} = \left(I - \frac{\partial f_\theta}{\partial x^*}\right)^{-1} \frac{\partial f_\theta}{\partial \theta}.
\tag{16}
$$

The chain rule implies $\frac{\partial L}{\partial \theta} = \frac{\partial L}{\partial x^*} \frac{\partial x^*}{\partial \theta}$. Substituting the expression for $\frac{\partial x^*}{\partial \theta}$, we obtain:

$$
\frac{\partial L}{\partial \theta} = \frac{\partial L}{\partial x^*} \left(I - \frac{\partial f_\theta}{\partial x^*}\right)^{-1} \frac{\partial f_\theta}{\partial \theta}.
\tag{17}
$$

$\square$

# B. Experimental Settings

We detail the benchmark in Appendix B.1, the baselines in Appendix B.2, and our implementation details in Appendix B.3.

## B.1. Benchmark

**Environment.** We adopt the robotic planning benchmark VirtualHome-Env (Liao et al., 2019) based on VirtualHome (Puig et al., 2018). It consists of a complex set of 292 planning tasks in 7 different indoor scenes, provided with 1360 mid-level action trajectories as ground truth annotations. These action trajectories are typically very long, with an average execution length of 10.8, highlighting its long-horizon characteristic. Moreover, the VirtualHome environment provides detailed feedback after performing each mid-level action, making it an ideal testbed for closed-loop planning.

Figure 7 visualizes a few examples sampled from the VirtualHome-Env benchmark. In each example, the planner is placed in an environment that spans a few indoor rooms and is given a detailed description of the environment. The description is originally in the form of a scene graph with objects as nodes and spatial relationships as edges, but for simplicity we present the planner with only the object nodes. The planner is then asked to generate a semantic action sequence based on a short task description. For instance, after receiving an instruction "turn on TV . . . ", the robot agent must first walk to the table and grab the remote control, and then point at the TV to turn it on. As seen, these planning tasks usually involve a rather complex scene setup, and the ground truth action sequences are quite long. We provide more detailed statistics in Figure 8.

Compared to alternative embodied planning benchmarks, VirtualHome-Env features both long time horizons and closed-loop feedback. For example, ALFRED (Shridhar et al., 2020) and ReALFRED (Kim et al., 2024) are two common embodied instruction following benchmarks, but their plan lengths are relatively short and can be determined by a few templates, making them unsuitable for long-horizon planning. PlanBench (Valmeekam et al., 2023) and TravelPlanner (Xie et al., 2024) are recent benchmarks designed specifically for LLM planning, but they do not provide closed-loop feedback during execution, which is an essential element of robotic planning. Therefore, we adopt VirtualHome-Env during the experiments.

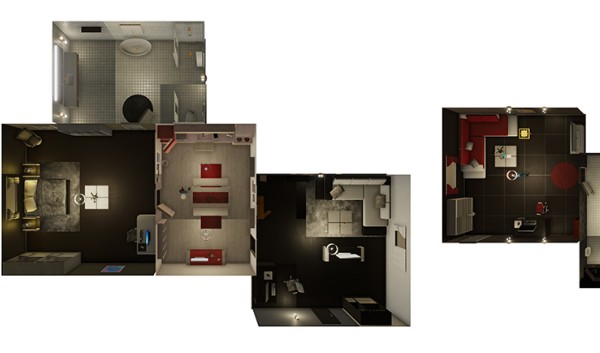 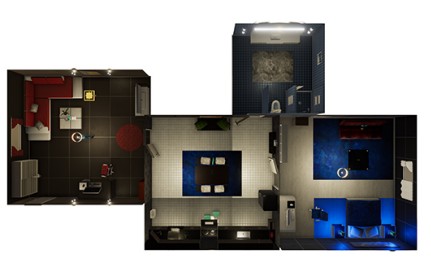 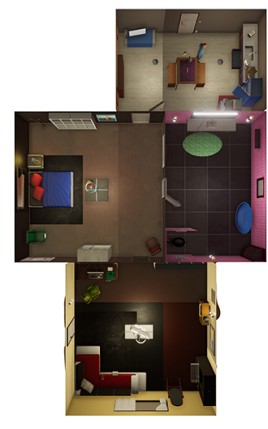

```
Task description:                Task description:                Task description:
Bring me red cookbook            Bring dirty plate to sink        Turn on TV with remote

Ground truth:                    Ground truth:                    Ground truth:
[WALK] <home_office>             [WALK] <dining_room>             [WALK] <home_office>
[WALK] <bookshelf>               [WALK] <table>                   [WALK] <table>
[FIND] <novel>                   [FIND] <table>                   [FIND] <remote_control>
[GRAB] <novel>                   [TURNTO] <table>                 [GRAB] <remote_control>
[WALK] <table>                   [FIND] <plate>                   [FIND] <television>
[PUTBACK] <novel> <table>        [GRAB] <plate>                   [TURNTO] <television>
                                 [WALK] <dining_room>             [POINTAT] <television>
                                 [WALK] <sink>                    [SWITCHON] <television>
                                 [FIND] <sink>                    [PUTOBJBACK] <remote_control>
                                 [PUTBACK] <plate> <sink>
```

|       (a)       |       (b)       |       (c)       |

Figure 7: Examples in VirtualHome-Env (Puig et al., 2018; Liao et al., 2019). The planner is given a detailed description of the environment (specifically, the objects within each rooms), a short task instruction, and is asked to output a sequence of mid-level actions associated with the correct objects.

**Action.** The VirtualHome environment (Puig et al., 2018) originally supported animating 12 atomic actions based on the Unity simulator, with the followup work VirtualHome-Env (Liao et al., 2019) adding support for more actions using a graph simulator. It currently supports 40 atomic actions, in which 21 actions can be animated through Unity. Each action is defined by an action name and some object arguments, and is implemented by prewritten code executors. In our experiments, we use a full set of 40 actions included in the VirtualHome-Env dataset, summarized as follows:

1. Actions without object association: SLEEP, STANDUP, WAKEUP.
2. Actions associated with one object: WALK, FIND, GRAB, WASH, WIPE, PULL, PUSH, POUR, TURNTO, POINTAT, WATCH, TOUCH, OPEN, CLOSE, RUN, SIT, READ, PUTON, PUTOFF, DROP, LIE, SWITCHON, SWITCHOFF, DRINK, LOOKAT, TYPE, CUT, PUTOBJBACK, EAT, RINSE, PLUGIN, PLUGOUT, GREET, SCRUB, SQUEEZE.
3. Actions associated with two objects: PUTIN, PUTBACK.

**Feedback.** Because the environment includes a graph simulator of the scene graph, it can respond quickly to actions, *e.g.* changing object attributes, and provide the updated scene graph at each step. In our experiments, we curate several types of closed-loop feedback based on these scene graphs, simulating coarse feedback that may be received in real-world situations. Specifically, we consider the following four categories of environmental feedback associated with task failure:

1. Program format feedback:"Your output does not conform to the required format", indicating that the generated action sequence does not conform to the required format.
2. Invalid command feedback: "Your output has an invalid command: ...", indicating that the generated action sequence has an illegal command line.
3. Execution feedback: "Your output is executed incorrectly in the environment.", indicating that the generated action sequence cannot be executed in the environment.
4. Task completion feedback: "You have not completed this task. The following objects and corresponding states do not meet the goals: ... The following objects have wrong relative position: ...", indicating that the generated action sequence cannot complete the task, with more details about the task failure.

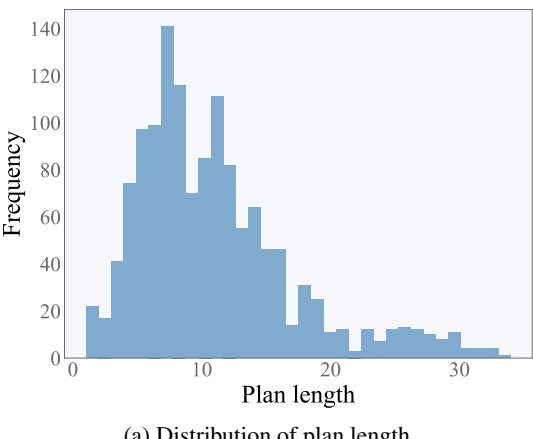

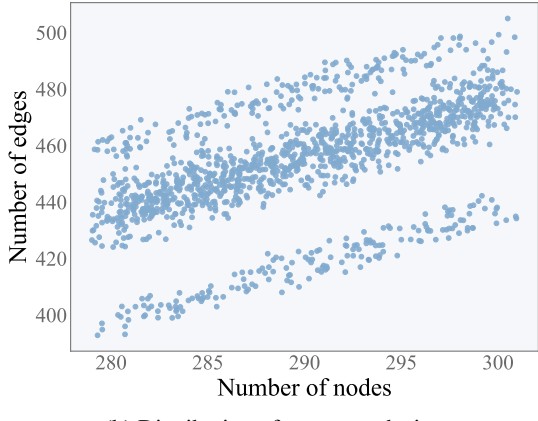

(a) Distribution of plan length

(b) Distribution of scene graph size

Figure 8: Detailed statistics of VirtualHome-Env (Puig et al., 2018; Liao et al., 2019). It features (a) a large set of long-horizon plans with an average length of 10.8, and (b) 7 complex scenes containing more than 280 objects and more than 400 valid relations. For clarity, we exclude the CLOSE and FACING relations, which are redundant for most planning tasks.

Table 6: Comparison of dataset protocols of VirtualHome (Puig et al., 2018). Since previous works have not released their dataset, we use the original VirtualHome-Env dataset (Liao et al., 2019) and perform our own partitioning.

|  | Public | #Tasks | #Scenes | Task Content | Task Splits |
|---|---|---|---|---|---|
| Tree-Planner |  | 35 | 4 | Household | Novel scene and task |
| LLM-MCTS |  | 2000 | 4 | Rearrangement | Novel scene and task, Novel scene, Novel task, Seen scene and task |
| Ours | ✓ | 1360 | 7 | Household | Novel scene and task, Novel scene, Novel task |

**Dataset Split.** We randomly divide the VirtualHome-Env dataset into training set and test set in a 50:50 ratio. To analyze the generalizability of our method, we mainly study the following three subsets of the test set: novel scene set, novel task set, and novel scene and task set. For instance, the novel scene set consists of seen planning tasks on unseen scenes. Overall, the dataset contains 735 training trajectories, 468 trajectories within the novel task set, 95 trajectories within the novel scene set, 62 trajectories within the novel scene and task set. Our models are first trained on the training set for a fixed number of epochs and then evaluated on the three test subsets above.

Note that there are two alternative dataset protocols for VirtualHome, represented by LLM-MCTS (Zhao et al., 2023) and Tree-Planne (Hu et al., 2024), as summarized in Table 6. Specifically, LLM-MCTS uses the indoor scenes of VirtualHome and synthesizes its own dataset focusing on object rearrangement tasks. Tree-Planner uses a subset of VirtualHome-Env to evaluate for training-free methods. However, since neither works has released their detailed dataset, therefore we simply use the original VirtualHome-Env dataset and partition it accordingly in our experiments.

### B.2. Baselines

Our method is mainly compared with Tree-Planner (Hu et al., 2024) and SELF-REFINE (Madaan et al., 2023), both reproduced using Llama 3 8B Instruct (Dubey et al., 2024) in line with ours. The former traverses an action tree that is built by repeated plan sampling, while the latter relies on self-refinement. To reproduce them, we perform supervised finetuning of Llama 3 on the training split of VirtualHome-Env for the same number of epochs as our method, and then follow their original procedures for inference. For instance, Tree-Planner is reproduced with both settings $N \in \{25, 50\}$ in action tree construction. The system prompts they use are similar to ours in Figure 9.

We also report the results summarized by Hu et al. (2024) for reference. They additionally considered ProgPrompt (Singh et al., 2023), Zero-shot Planner (Huang et al., 2022) and two self-refinement planners, Local Replan (Raman et al., 2022; Guo et al., 2023) and Global Replan (Shinn et al., 2023). Since these baselines were implemented by calling the GPT-3.5 API instead of finetuning Llama 3, we report them in the novel scene and task track for a relatively fair comparison. It is worth noting that they adopted a smaller subset of actions and feedback, and differed in the curation of partial observations of the environment. Therefore, their results are presented for reference only.

There are several robotic planning baselines that we have not compared due to large environmental differences. For example, LLM-MCTS (Zhao et al., 2023) is a representative tree-search (Yao et al., 2023a) based planner. It followed Watch-and-help (Puig et al., 2021) to generate a dataset of simple embodied tasks (mostly object rearrangement tasks), while our work considers a more complex set of planning tasks, see Table 6. Alternative planners based on symbolic scene graph (Zhu et al., 2021; Rana et al., 2023), code (Liang et al., 2023; Sun et al., 2023), or PDDL (McDermott, 2000; Liu et al., 2023; Guan et al., 2023) are less flexible and difficult to implement in our environment.

System Prompt

```
You need to act as a task planner, who first draft an initial sub-task sequence and then refine it in the next few
    iterations.
When the the draft sub-task sequence is Null, you should output the initial sub-task sequence.
When the the draft sub-task sequence is not Null, You should refine it based on the the draft sub-task sequence.
If you have previously generated some action sequences and tried to execute them in the environment, their feedback will be
    provided to you for reference.
Each sub-task can be one of the following form: 1. [action_name]; 2. [action_name] <object name 1> (object id 1); 3. [
    action_name] <object name 1> (object id 1) <object name 2> (object id 2).
The (object id) is used to tell the simulator which object the action should act on.
The number of arguments depends on the action type.
For action type 1, the available actions are: SLEEP, STANDUP, WAKEUP
For action type 2, the available actions are: WALK, FIND, GRAB, WASH, WIPE, PULL, PUSH, POUR, TURNTO, POINTAT, WATCH, TOUCH
    , OPEN, CLOSE, RUN, SIT, READ, PUTON, PUTOFF, DROP, LIE, SWITCHON, SWITCHOFF, DRINK, LOOKAT, TYPE, CUT, PUTOBJBACK,
    EAT, RINSE, PLUGIN, PLUGOUT, GREET, SCRUB, SQUEEZE
For action type 3, the available actions are: PUTIN, PUTBACK
All action_name of the sub-tasks must be chosen from the above actions.
You should output the sub-task sequence in succinct form.
You must output END after you have output the entire sub-task sequence.
```

Task

```
Task name:
Grab some juice

Instructions:
I go to the fridge, and grab some juice out of it. I then get a glass, and pour the juice into the glass.
```

Env

```
There are 4 rooms, and you are an embodied character with ID 198 in bedroom with ID 199.
The objects in each room is as follows:

Room name: home_office
Room ID: 1
Object ID and name in this room:
28 hanger
73 mat
......
Room name: dining_room
Room ID: 100
Object ID and name in this room:
116 ceiling
2005 food_food
......
```

Feedback $c_t$

```
Feedbacks from past executions:
Action sequence:
[WALK] <dining_room> (100)
[WALK] <cupboard> (132)
[FIND] <cupboard> (132)
[OPEN] <cupboard> (132)
[FIND] <cup> (1000)
[GRAB] <cup> (1000)
[CLOSE] <cupboard> (132)
[WALK] <freezer> (141)
[OPEN] <freezer> (141)
[FIND] <juice> (1001)
[GRAB] <juice> (1001)
[POUR] <juice> (1001) <cup> (1000)
[PUTOBJBACK] <juice> (1001)
[CLOSE] <freezer> (141)
[END]
Feedback:
You have not completed this task.
The following objects have wrong relative position: (1000, cup) and (128, table).
```

Draft Plan $x_t$

```
The draft sub-task sequence:
[WALK] <dining_room> (100)
[WALK] <cupboard> (132)
[FIND] <cupboard> (132)
[OPEN] <cupboard> (132)
[FIND] <cup> (1000)
[GRAB] <cup> (1000)
[CLOSE] <cupboard> (132)
[WALK] <freezer> (141)
[OPEN] <freezer> (141)
[FIND] <juice> (1001)
[GRAB] <juice> (1001)
[POUR] <juice> (1001) <cup> (1000)
[PUTOBJBACK] <juice> (1001)
[CLOSE] <freezer> (141)
[END]
```

Figure 9: Example of the prompt used by our equilibrium planner.

### B.3. Implementation Details

**Prompt.** Our approach involves two LLMs, one LLM as the equilibrium planner and an additional LLM as an optional world model. The planner's input is illustrated in Figure 9. As can be seen, it consists of five parts: system prompt, task definition, environment description, history feedback, and draft plan. Notably, the system prompt is modified from Hu et al. (2024), and the environment section describes the initial environment sorted by rooms, including the object names with their IDs within each room. For the optional world model, we adopt a similar prompt, except that it receives more information about the initial environment, including edges in the scene graph that indicate spatial relations. This additional information helps the world model to better predict the environmental feedback given a generated plan.

**Finetuning.** Both our equilibrium planner and the world model are finetuned in a supervised manner. The equilibrium planner is finetuned for 6 iterations with a learning rate of 0.0002. The training data is constructed adaptively using all previous equilibrium solutions. Specifically, an equilibrium memory is maintained that buffers all equilibrium solutions, including the newest ones. At each iteration, we curate the training data by weighted sampling from this memory (where the newest solutions are sampled more frequently) and then pairing them with the ground truths. To prevent overfitting to the history equilibrium, a decay ratio of 0.5 is used when sampling from the fixed points of previous iterations. Thereafter, we update the model parameters using gradient descent according to Equation (6) for one epoch per iteration.

For the world model, we collect all interacting experiences between the planner and the environment, including plans and feedback, and finetune it for 5 epochs using the same learning rate of 0.0002. The world model is initialized from Llama 3 8B Instruct and supervised finetuned on all the planner's environmental interactions. This procedure takes place after the equilibrium planner has completed training, so that all of its data can be leveraged at once. The finetuning data is constructed in a format similar to Figure 9 in the appendix, with a different order to predict feedback from the plan. Finetuning the world model takes about 30 hours due to its longer context (e.g. spatial relations).

**Inference.** The inference of our planner is described in Algorithm 1, which involves a nested equilibrium solving process. Given the environment and the task instruction, we initialize the draft plan $x_0$ and the feedback $c_0$ as null and iterate through a nested loop. Each inner loop reuses the feedback from the outer loop to self-refine the draft plan, and after the inner loop converges, we update the feedback by interacting with the environment or world model. The ratio of environmental interactions to world model calls is currently set to 1:1, *i.e.*, the planner alternates between using the environmental feedback and the world model at each loop. Note that it is possible to reduce this ratio and the number of environmental interactions required if the planner has access to a more accurate world model. This process continues until it converges to an equilibrium point or reaches an upper bound on the outer loop, which we set to 10 to match Tree-Planner (Hu et al., 2024) but is rarely reached. Thus, the inference compute used is mostly determined by the convergence speed of the model itself.

Our LLM sampling strategy facilitates model convergence. Specifically, we use greedy sampling to stabilize LLM outputs, except that the first refinement step uses top-k sampling with $k = 10$ for higher diversity. Since most of the text prompt remains unchanged during equilibrium solving, we employ KV cache to accelerate inference, which can be further improved with parallel decoding techniques (Santilli et al., 2023; Cai et al., 2024; Kou et al., 2024).

## C. Additional Results

**Effectiveness of our full method.** We exemplify the self-correction trajectories of our full method in Figs. 10 and 11. Compared to SELF-REFINE (Madaan et al., 2023) and Tree-Planner (Hu et al., 2024), our approach is more competent in revising a long plan through few forward passes without additional system 2. This is attributed to our efficient training scheme for teaching planners to self-refine. We also compare different types of feedback utilized by our planner in Figure 12. As can be observed, internal feedback alone cannot enable successful replanning, but it can reduce environmental interactions prior to convergence. This confirms the effectiveness of both internal and external feedback in closed-loop planning.

**Effectiveness of the inner loop.** Table 8 illustrates that even without feedback, our method yields significant performance improvements. To better understand this, we visualize the self-refining traces of the inner loop in Figures 13 and 14. Specifically, Figure 13 shows the inference trace at each inner-loop iteration. Even though the planner only succeeded in later steps, there are consistent quality improvements in its output during the inner-loop introspection without any feedback. Figure 14 compares our inference trace to a prompting-based alternative, where our method shows to be more effective at steering the output toward a correct plan via the inner loop. The working mechanism of our inner loops was mentioned in the introduction, *i.e.* allowing for bidirectional dependency as well as scaling of inference-time compute.

Table 8: Effectiveness of our method in the no-feedback setting, where it shows clear performance advantages.

| | Novel Scene and Task | | | Novel Scene | | | Novel Task | | |
|---|---|---|---|---|---|---|---|---|---|
| | Exec. | SR | GCR | Exec. | SR | GCR | Exec. | SR | GCR |
| Supervised | 93.55 | 24.19 | 32.55 | 96.84 | 41.05 | 49.81 | 95.94 | 26.07 | 35.53 |
| SELF-REFINE | 72.58 | 32.26 | 52.29 | 74.74 | 44.21 | 62.25 | 65.38 | 30.98 | 51.80 |
| Ours | 88.71 | **33.87** | **59.98** | 96.79 | **49.47** | **66.60** | 93.80 | **34.62** | **59.06** |

Table 9: Comparison to Tree-Planner with and without world model. Only our method shows a significant improvement.

| | World model | Novel Scene and Task | | | Novel Scene | | | Novel Task | | |
|---|---|---|---|---|---|---|---|---|---|---|
| | | Exec. | SR | GCR | Exec. | SR | GCR | Exec. | SR | GCR |
| Tree-Planner$_{N=25}$ | | 95.16 | 38.71 | 63.18 | 96.08 | 51.58 | 69.45 | 95.50 | 40.38 | **63.75** |
| Tree-Planner$_{N=25}$ | ✓ | 96.25 | 38.71 | 58.71 | 98.81 | 51.58 | 63.94 | 96.66 | 38.46 | 57.40 |
| Tree-Planner$_{N=50}$ | | 94.94 | 38.71 | 63.50 | 96.06 | 51.58 | 69.54 | 95.40 | 39.74 | 63.29 |
| Tree-Planner$_{N=50}$ | ✓ | 96.16 | 37.10 | 57.53 | 98.22 | 54.74 | 69.64 | 96.80 | 39.32 | 58.84 |
| Ours | | 88.71 | 33.87 | 59.98 | 96.79 | 49.47 | 66.60 | 93.80 | 34.62 | 59.06 |
| Ours | ✓ | 90.32 | **40.32** | **65.40** | 95.79 | **65.26** | **79.47** | 93.38 | **41.88** | 62.76 |

**Comparison with Tree-Planner.** As shown in Table 9, our performance is inferior to Tree-Planner in the no-feedback setting because of the difference in refining a single plan with <10 iterations instead of generating 25 or 50 candidate plans (giving Tree-Planner a comparative advantage). However, in the other settings that allow feedback, our method outperforms Tree-Planner by incorporating feedback more flexibly. By taking into account feedback through forward passes of LLMs, our method allows arbitrary changes based on the LLMs' knowledge, and correcting multiple errors in parallel (Figure 10c). In contrast, tree-based alternatives require backtracking in a tree, which is costly when correcting an early mistake and does not fully exploit the implicit knowledge in feedback. For example in Figure 11, Tree-Planner ignored the earlier feedback and made the same mistake twice ([OPEN] ⟨laptop⟩).

**Generalization to environment without feedback.** we test our pre-trained VirtualHome planner on ALFRED benchmark with updated prompts. For evaluation, we consider two planning metrics: task classification accuracy (across 7 task types in ALFRED) and recall of ground-truth action-object pairs in the predicted plan. As shown in Table 7, our planner generalizes significantly better than the supervised trained planner.

Table 7: Zero-shot evaluation on ALFRED. Our model demonstrates better generalization without retraining.

| | Task classification acc. | Action-object recall |
|---|---|---|
| SFT Llama | 11% | 0.50% |
| Ours | **54%** | **27.08%** |

# D. Limitations

While our equilibrium sequence modeling improves the planning capability of LLMs, we identify the following failure scenarios during the experiments: (1) hallucination of the equilibrium planner and the world model as in vanilla LLMs; (2) lack of awareness of history context such as previously grabbed objects. The latter can be resolved with the context module in Kim et al. (2023a) or reasoning techniques as in Yao et al. (2023b).

In a broader sense, our method may be limited in generalizing to new domains because it requires the ground truth and environmental feedback during training. These procedures with the equilibrium solving process results in lower training efficiency. Also, the current formulation only considers the explicit output plan without implicit reasoning steps. While it is possible to synergize planning and reasoning with their combined advances, we leave this to future work due to feasibility constraints for small-scale experiments with large reasoning LLMs (Jaech et al., 2024; Guo et al., 2025). Furthermore, our model has only text input and no visual input, which limits its applicability in the real world. This can be resolved by introducing video-based planners (Du et al., 2024), world models (Yang et al., 2024a; Brooks et al., 2024) and vision-language models(Yang et al., 2024b; Athalye et al., 2024).

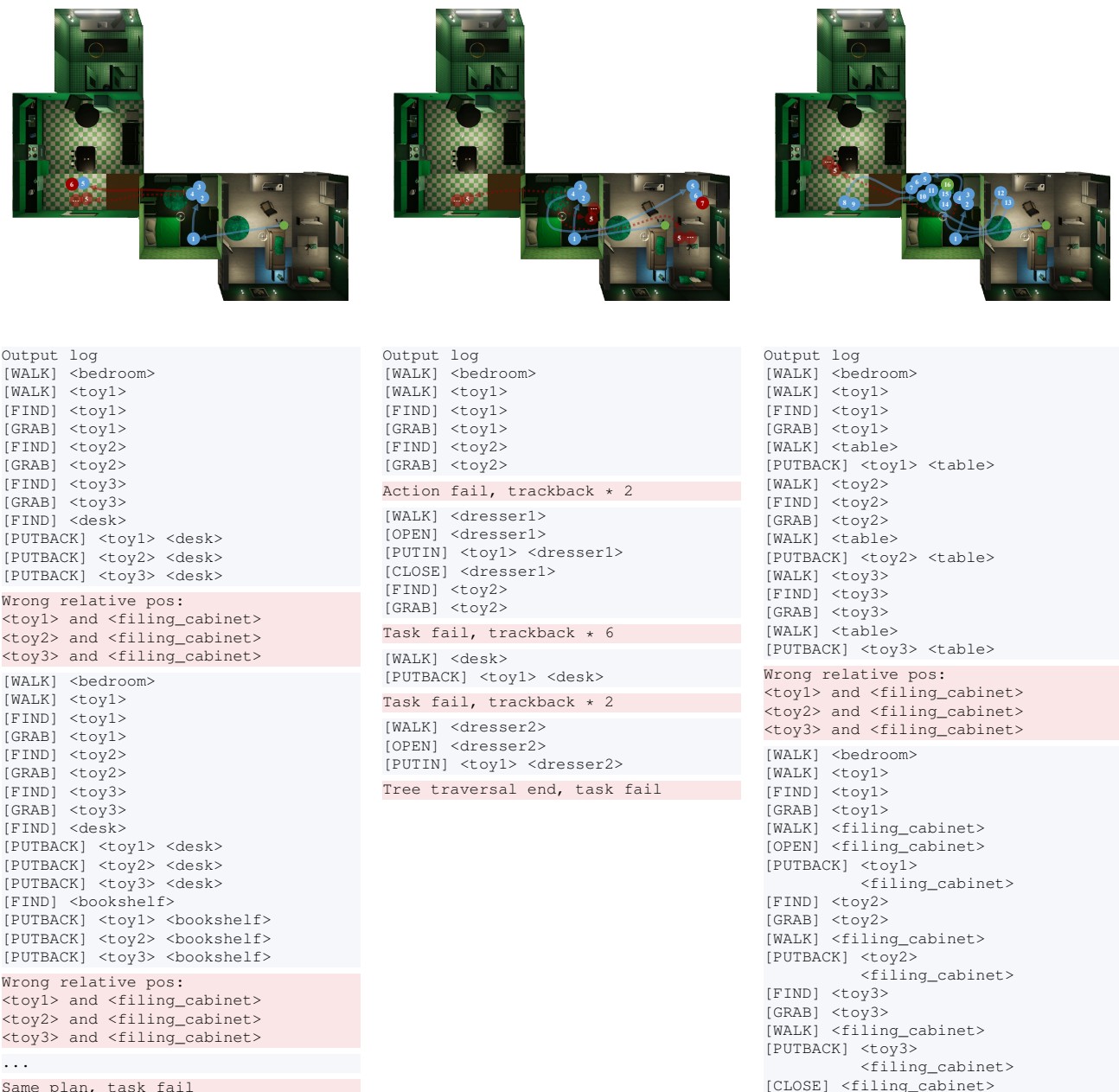

```
Output log
[WALK] <bedroom>
[WALK] <toy1>
[FIND] <toy1>
[GRAB] <toy1>
[FIND] <toy2>
[GRAB] <toy2>
[FIND] <toy3>
[GRAB] <toy3>
[FIND] <desk>
[PUTBACK] <toy1> <desk>
[PUTBACK] <toy2> <desk>
[PUTBACK] <toy3> <desk>

Wrong relative pos:
<toy1> and <filing_cabinet>
<toy2> and <filing_cabinet>
<toy3> and <filing_cabinet>

[WALK] <bedroom>
[WALK] <toy1>
[FIND] <toy1>
[GRAB] <toy1>
[FIND] <toy2>
[GRAB] <toy2>
[FIND] <toy3>
[GRAB] <toy3>
[FIND] <desk>
[PUTBACK] <toy1> <desk>
[PUTBACK] <toy2> <desk>
[PUTBACK] <toy3> <desk>
[FIND] <bookshelf>
[PUTBACK] <toy1> <bookshelf>
[PUTBACK] <toy2> <bookshelf>
[PUTBACK] <toy3> <bookshelf>

Wrong relative pos:
<toy1> and <filing_cabinet>
<toy2> and <filing_cabinet>
<toy3> and <filing_cabinet>

...

Same plan, task fail
```

(a) SELF-REFINE

```
Output log
[WALK] <bedroom>
[WALK] <toy1>
[FIND] <toy1>
[GRAB] <toy1>
[FIND] <toy2>
[GRAB] <toy2>

Action fail, trackback * 2

[WALK] <dresser1>
[OPEN] <dresser1>
[PUTIN] <toy1> <dresser1>
[CLOSE] <dresser1>
[FIND] <toy2>
[GRAB] <toy2>

Task fail, trackback * 6

[WALK] <desk>
[PUTBACK] <toy1> <desk>

Task fail, trackback * 2

[WALK] <dresser2>
[OPEN] <dresser2>
[PUTIN] <toy1> <dresser2>

Tree traversal end, task fail
```

(b) Tree-Planner

```
Output log
[WALK] <bedroom>
[WALK] <toy1>
[FIND] <toy1>
[GRAB] <toy1>
[WALK] <table>
[PUTBACK] <toy1> <table>
[WALK] <toy2>
[FIND] <toy2>
[GRAB] <toy2>
[WALK] <table>
[PUTBACK] <toy2> <table>
[WALK] <toy3>
[FIND] <toy3>
[GRAB] <toy3>
[WALK] <table>
[PUTBACK] <toy3> <table>

Wrong relative pos:
<toy1> and <filing_cabinet>
<toy2> and <filing_cabinet>
<toy3> and <filing_cabinet>

[WALK] <bedroom>
[WALK] <toy1>
[FIND] <toy1>
[GRAB] <toy1>
[WALK] <filing_cabinet>
[OPEN] <filing_cabinet>
[PUTBACK] <toy1>
          <filing_cabinet>
[FIND] <toy2>
[GRAB] <toy2>
[WALK] <filing_cabinet>
[PUTBACK] <toy2>
          <filing_cabinet>
[FIND] <toy3>
[GRAB] <toy3>
[WALK] <filing_cabinet>
[PUTBACK] <toy3>
          <filing_cabinet>
[CLOSE] <filing_cabinet>

Task success
```

(c) Ours

Figure 10: Visualization of our self-correction process in comparison with baselines. This example uses only environmental feedback, and we include toy IDs in the presentation for clarity. The task instruction is "Pick up all the toys on the floor and put them in their correct storage bin or shelf".

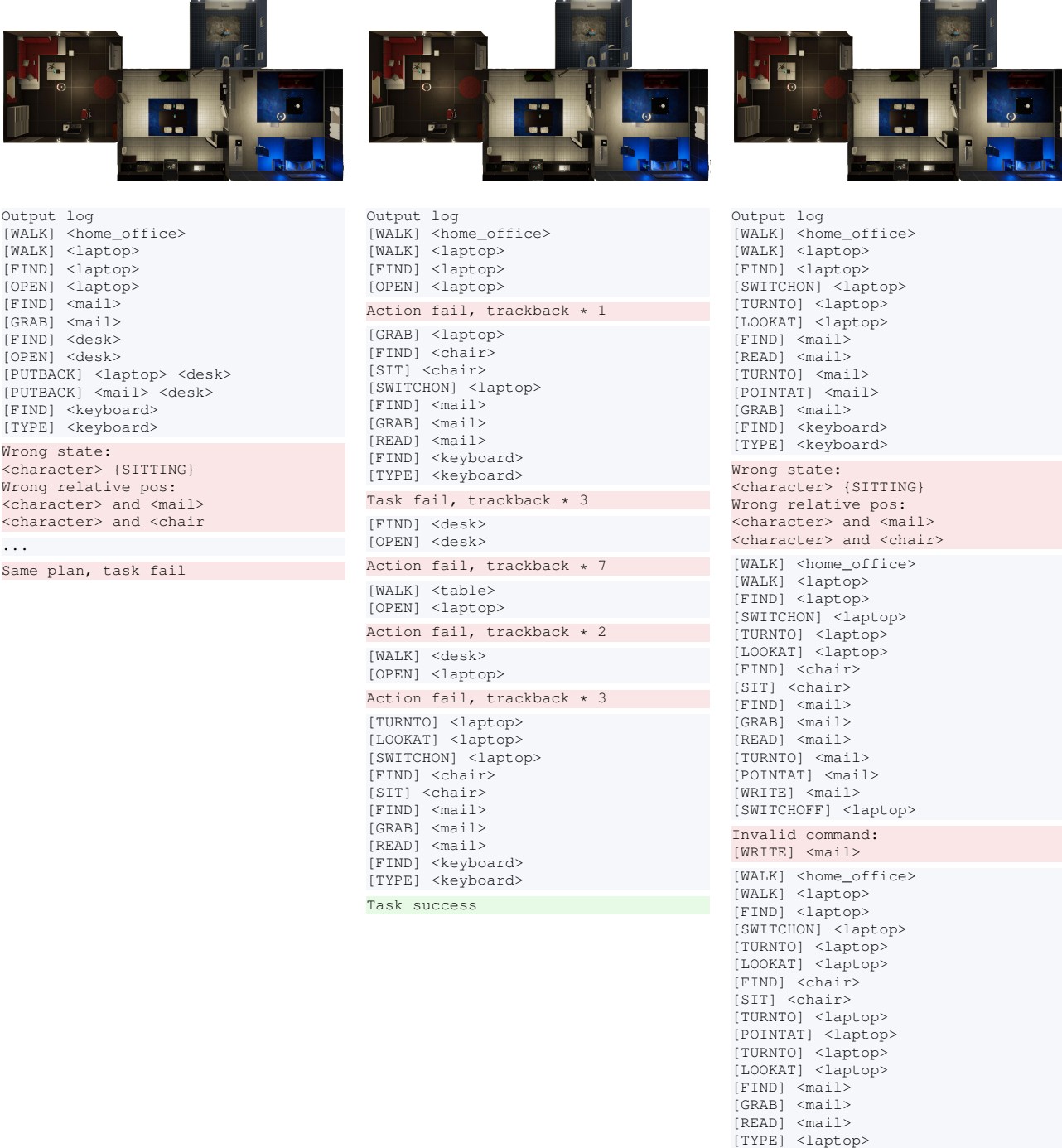

```
Output log
[WALK] <home_office>
[WALK] <laptop>
[FIND] <laptop>
[OPEN] <laptop>
[FIND] <mail>
[GRAB] <mail>
[FIND] <desk>
[OPEN] <desk>
[PUTBACK] <laptop> <desk>
[PUTBACK] <mail> <desk>
[FIND] <keyboard>
[TYPE] <keyboard>

Wrong state:
<character> {SITTING}
Wrong relative pos:
<character> and <mail>
<character> and <chair>

...

Same plan, task fail
```

(a) SELF-REFINE

```
Output log
[WALK] <home_office>
[WALK] <laptop>
[FIND] <laptop>
[OPEN] <laptop>

Action fail, trackback * 1

[GRAB] <laptop>
[FIND] <chair>
[SIT] <chair>
[SWITCHON] <laptop>
[FIND] <mail>
[GRAB] <mail>
[READ] <mail>
[FIND] <keyboard>
[TYPE] <keyboard>

Task fail, trackback * 3

[FIND] <desk>
[OPEN] <desk>

Action fail, trackback * 7

[WALK] <table>
[OPEN] <laptop>

Action fail, trackback * 2

[WALK] <desk>
[OPEN] <laptop>

Action fail, trackback * 3

[TURNTO] <laptop>
[LOOKAT] <laptop>
[SWITCHON] <laptop>
[FIND] <chair>
[SIT] <chair>
[FIND] <mail>
[GRAB] <mail>
[READ] <mail>
[FIND] <keyboard>
[TYPE] <keyboard>

Task success
```

(b) Tree-Planner

```
Output log
[WALK] <home_office>
[WALK] <laptop>
[FIND] <laptop>
[SWITCHON] <laptop>
[TURNTO] <laptop>
[LOOKAT] <laptop>
[FIND] <mail>
[READ] <mail>
[TURNTO] <mail>
[POINTAT] <mail>
[GRAB] <mail>
[FIND] <keyboard>
[TYPE] <keyboard>

Wrong state:
<character> {SITTING}
Wrong relative pos:
<character> and <mail>
<character> and <chair>

[WALK] <home_office>
[WALK] <laptop>
[FIND] <laptop>
[SWITCHON] <laptop>
[TURNTO] <laptop>
[LOOKAT] <laptop>
[FIND] <chair>
[SIT] <chair>
[FIND] <mail>
[GRAB] <mail>
[READ] <mail>
[TURNTO] <mail>
[POINTAT] <mail>
[WRITE] <mail>
[SWITCHOFF] <laptop>

Invalid command:
[WRITE] <mail>

[WALK] <home_office>
[WALK] <laptop>
[FIND] <laptop>
[SWITCHON] <laptop>
[TURNTO] <laptop>
[LOOKAT] <laptop>
[FIND] <chair>
[SIT] <chair>
[TURNTO] <laptop>
[POINTAT] <laptop>
[TURNTO] <laptop>
[LOOKAT] <laptop>
[FIND] <mail>
[GRAB] <mail>
[READ] <mail>
[TYPE] <laptop>

Task success
```

(c) Ours

Figure 11: Visualization of our self-correction process in comparison with baselines. This example uses only environmental feedback. The task instruction is "Open email application, open new emails and respond accordingly". Our proposed method succeeds with fewer external feedback.

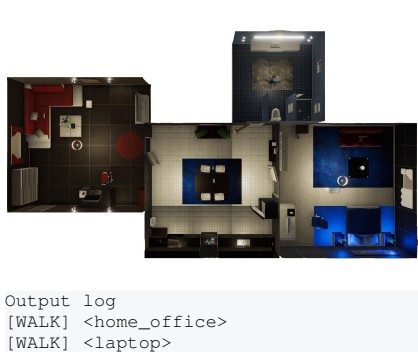
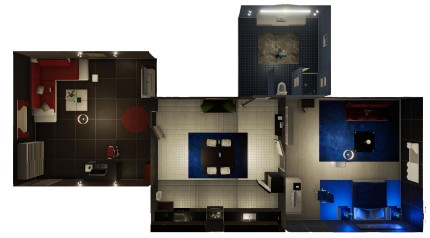
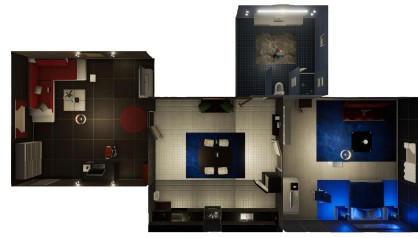

```
Output log
[WALK] <home_office>
[WALK] <laptop>
[FIND] <laptop>
[SWITCHON] <laptop>
[TURNTO] <laptop>
[LOOKAT] <laptop>
[FIND] <mail>
[READ] <mail>
[TURNTO] <mail>
[POINTAT] <mail>
[GRAB] <mail>
[FIND] <keyboard>
[TYPE] <keyboard>
```
```
World model feedback
Wrong state:
<character> {SITTING}
<computer> {ON}
<computer> {PLUGGED_IN}
Wrong relative pos:
<character> and <chair>
```
```
[WALK] <home_office>
[WALK] <chair>
[FIND] <chair>
[SIT] <chair>
[FIND] <computer>
[SWITCHON] <computer>
[TURNTO] <computer>
[LOOKAT] <computer>
[FIND] <laptop>
[TURNTO] <laptop>
[LOOKAT] <laptop>
[FIND] <mail>
[TURNTO] <mail>
[LOOKAT] <mail>
[GRAB] <mail>
[READ] <mail>
[FIND] <keyboard>
[TYPE] <keyboard>
```
```
World model feedback
Task success
```
```
Task fail
```

```
Output log
[WALK] <home_office>
[WALK] <laptop>
[FIND] <laptop>
[SWITCHON] <laptop>
[TURNTO] <laptop>
[LOOKAT] <laptop>
[FIND] <mail>
[READ] <mail>
[TURNTO] <mail>
[POINTAT] <mail>
[GRAB] <mail>
[FIND] <keyboard>
[TYPE] <keyboard>
```
```
Environmental feedback
Wrong state:
<character> {SITTING}
Wrong relative pos:
<character> and <mail>
<character> and <chair>
```
```
[WALK] <home_office>
[WALK] <laptop>
[FIND] <laptop>
[SWITCHON] <laptop>
[TURNTO] <laptop>
[LOOKAT] <laptop>
[FIND] <chair>
[SIT] <chair>
[FIND] <mail>
[GRAB] <mail>
[READ] <mail>
[TURNTO] <mail>
[POINTAT] <mail>
[WRITE] <mail>
[SWITCHOFF] <laptop>
```
```
Environmental feedback
Invalid command:
[WRITE] <mail>
```
```
[WALK] <home_office>
[WALK] <laptop>
[FIND] <laptop>
[SWITCHON] <laptop>
[TURNTO] <laptop>
[LOOKAT] <laptop>
[FIND] <chair>
[SIT] <chair>
[TURNTO] <laptop>
[POINTAT] <laptop>
[TURNTO] <laptop>
[LOOKAT] <laptop>
[FIND] <mail>
[GRAB] <mail>
[READ] <mail>
[TYPE] <laptop>
```
```
Task success
```

```
Output log
[WALK] <home_office>
[WALK] <laptop>
[FIND] <laptop>
[SWITCHON] <laptop>
[TURNTO] <laptop>
[LOOKAT] <laptop>
[FIND] <mail>
[READ] <mail>
[TURNTO] <mail>
[POINTAT] <mail>
[GRAB] <mail>
[FIND] <keyboard>
[TYPE] <keyboard>
```
```
World model feedback
Wrong state:
<character> {SITTING}
<computer> {ON}
<computer> {PLUGGED_IN}
Wrong relative pos:
<character> and <chair>
```
```
[WALK] <home_office>
[WALK] <chair>
[FIND] <chair>
[SIT] <chair>
[FIND] <computer>
[SWITCHON] <computer>
[TURNTO] <computer>
[LOOKAT] <computer>
[FIND] <laptop>
[TURNTO] <laptop>
[LOOKAT] <laptop>
[FIND] <mail>
[TURNTO] <mail>
[LOOKAT] <mail>
[GRAB] <mail>
[READ] <mail>
[FIND] <keyboard>
[TYPE] <keyboard>
```
```
Environmental feedback
Wrong relative pos:
<character> and <mail>
```
```
[WALK] <home_office>
[WALK] <laptop>
[FIND] <laptop>
[SWITCHON] <laptop>
[TURNTO] <laptop>
[LOOKAT] <laptop>
[FIND] <chair>
[SIT] <chair>
[TURNTO] <laptop>
[POINTAT] <laptop>
[WATCH] <laptop>
[FIND] <mail>
[TURNTO] <mail>
[LOOKAT] <mail>
[GRAB] <mail>
[READ] <mail>
[TYPE] <laptop>
```
```
World model feedback
Task success
```
```
Task success
```

(a) Ours w/o external feedback      (b) Ours w/o internal feedback      (c) Ours

Figure 12: Visualization of our self-correction process with different types of feedback. The task instruction is "Open email application, open new emails and respond accordingly".

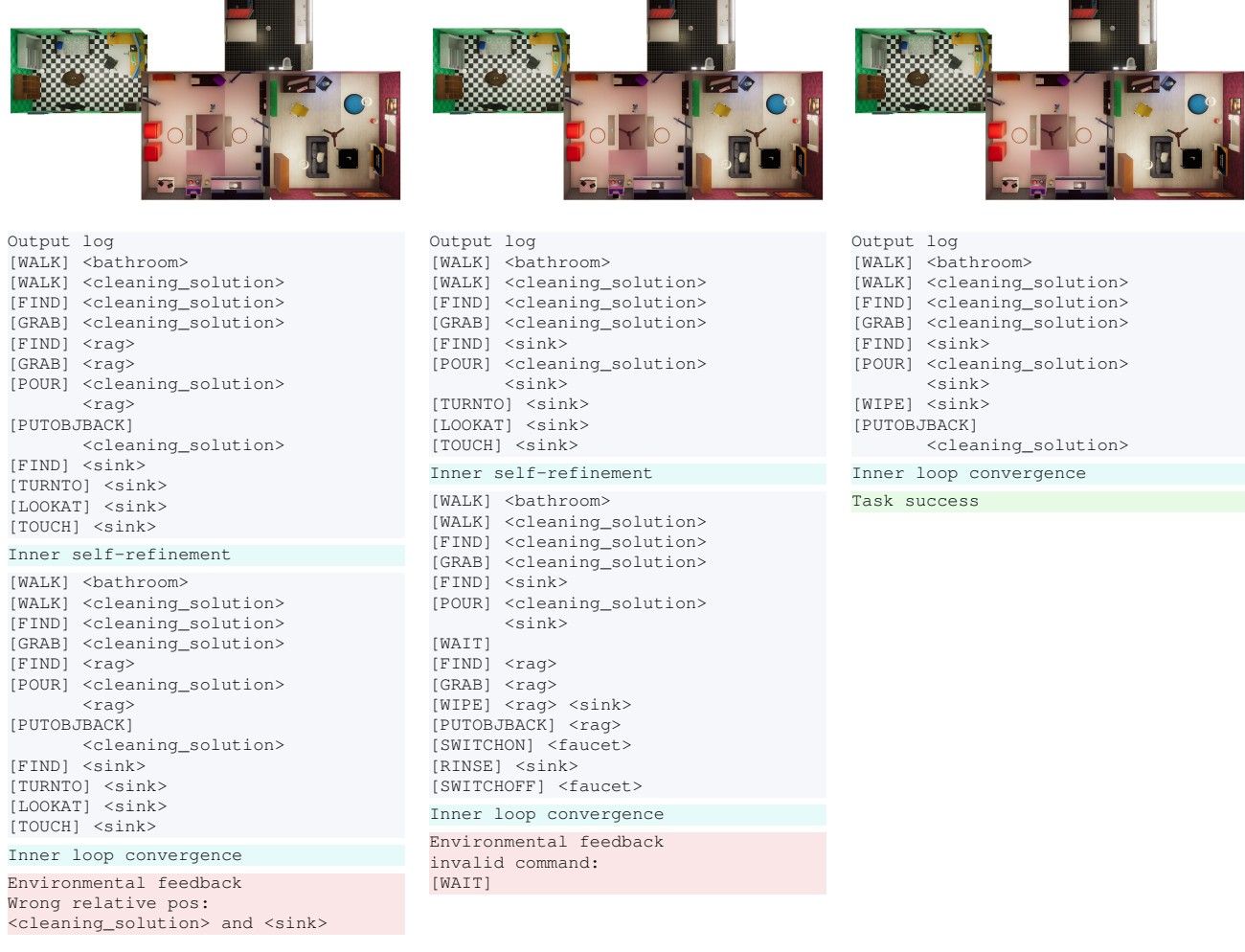

```
Output log
[WALK] <bathroom>
[WALK] <cleaning_solution>
[FIND] <cleaning_solution>
[GRAB] <cleaning_solution>
[FIND] <rag>
[GRAB] <rag>
[POUR] <cleaning_solution>
       <rag>
[PUTOBJBACK]
       <cleaning_solution>
[FIND] <sink>
[TURNTO] <sink>
[LOOKAT] <sink>
[TOUCH] <sink>

Inner self-refinement

[WALK] <bathroom>
[WALK] <cleaning_solution>
[FIND] <cleaning_solution>
[GRAB] <cleaning_solution>
[FIND] <rag>
[POUR] <cleaning_solution>
       <rag>
[PUTOBJBACK]
       <cleaning_solution>
[FIND] <sink>
[TURNTO] <sink>
[LOOKAT] <sink>
[TOUCH] <sink>

Inner loop convergence

Environmental feedback
Wrong relative pos:
<cleaning_solution> and <sink>
```

(a) The 1ˢᵗ inner loop

```
Output log
[WALK] <bathroom>
[WALK] <cleaning_solution>
[FIND] <cleaning_solution>
[GRAB] <cleaning_solution>
[FIND] <sink>
[POUR] <cleaning_solution>
       <sink>
[TURNTO] <sink>
[LOOKAT] <sink>
[TOUCH] <sink>

Inner self-refinement

[WALK] <bathroom>
[WALK] <cleaning_solution>
[FIND] <cleaning_solution>
[GRAB] <cleaning_solution>
[FIND] <sink>
[POUR] <cleaning_solution>
       <sink>
[WAIT]
[FIND] <rag>
[GRAB] <rag>
[WIPE] <rag> <sink>
[PUTOBJBACK] <rag>
[SWITCHON] <faucet>
[RINSE] <sink>
[SWITCHOFF] <faucet>

Inner loop convergence

Environmental feedback
invalid command:
[WAIT]
```

(b) The 2ⁿᵈ inner loop

```
Output log
[WALK] <bathroom>
[WALK] <cleaning_solution>
[FIND] <cleaning_solution>
[GRAB] <cleaning_solution>
[FIND] <sink>
[POUR] <cleaning_solution>
       <sink>
[WIPE] <sink>
[PUTOBJBACK]
       <cleaning_solution>

Inner loop convergence

Task success
```

(c) The 3ʳᵈ inner loop

Figure 13: Visualization of our self-correction process with inner-loop traces. This example uses only environmental feedback. The task instruction is "Spray some cleaner and wait a minute then wipe it out". As can be seen, each inner loop improves planning quality: the first removes a redundant [GRAB] action, while the second adds a crucial [WIPE] action and other interesting details.

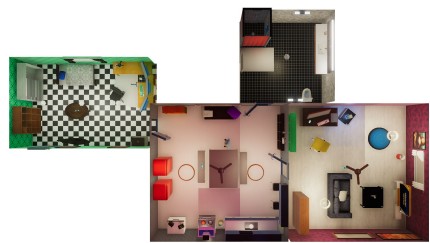

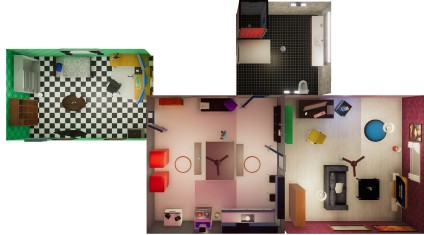

```
Output log
[WALK] <bathroom>
[WALK] <electric_shaver>
[FIND] <electric_shaver>
[GRAB] <electric_shaver>
[FIND] <mirror>
[TURNTO] <mirror>
[LOOKAT] <mirror>
[FIND] <chair>
[SIT] <chair>
[SWITCHON] <electric_shaver>
[FIND] <beard>
[TURNTO] <beard>
[POINTAT] <beard>
[FIND] <beard>
[WIPE] <beard>
```

Inner loop convergence

invalid command:
[FIND] <beard>
Task fail

```
Output log
[WALK] <bathroom>
[WALK] <electric_shaver>
[FIND] <electric_shaver>
[GRAB] <electric_shaver>
[FIND] <mirror>
[TURNTO] <mirror>
[LOOKAT] <mirror>
[SWITCHON] <electric_shaver>
[RUN] <mirror>
[TURNTO] <mirror>
[LOOKAT] <mirror>
[SWITCHOFF] <electric_shaver>
[PUTOBJBACK] <electric_shaver>
```

Inner self-refinement

```
[WALK] <bathroom>
[WALK] <electric_shaver>
[FIND] <electric_shaver>
[GRAB] <electric_shaver>
[FIND] <mirror>
[TURNTO] <mirror>
[LOOKAT] <mirror>
[SWITCHON] <electric_shaver>
[SWITCHOFF] <electric_shaver>
```

Inner loop convergence

Task success

(a) Prompt-based self-refinement                    (b) Ours

Figure 14: Visualization of our self-correction process in comparison with prompting-based method without any feedback. The task instruction is "Pick up razor and shave yourself". As can be seen, the prompting-based model converges without any self-correction, while our model achieves success.

