# OpenReview forum: "Closed-Loop Long-Horizon Robotic Planning via Equilibrium Sequence Modeling"
_ICML.cc/2025/Conference — ICML 2025 poster_

### Official Review · Reviewer_Bgnq · 2025-03-13

**Overall Recommendation:** 4

**Summary:**

This paper introduces a novel approach for long-horizon robotic task planning using LLMs with Equilibrium Sequence Modeling. The core idea is to view planning as a self-refinement process converging to a fixed-point (equilibrium) plan that integrates feedback from the environment or a world model. Rather than relying on reinforcement learning or complex prompting strategies, the paper presents an end-to-end supervised training framework grounded in deep equilibrium models. Additionally, it introduces a nested equilibrium modeling loop to efficiently incorporate closed-loop feedback. The proposed method is evaluated on the VirtualHome-Env benchmark and demonstrates state-of-the-art performance compared to Tree-Planner and SELF-REFINE approaches, particularly in success rate (SR) and goal condition recall (GCR).

**Claims And Evidence:**

The central claims are:
* Self-refinement can be formulated as an equilibrium sequence modeling problem.
* Such formulation enables efficient supervised training without reinforcement learning or verifiers.
* Nested equilibrium modeling improves closed-loop planning using minimal feedback.
* The approach scales better w.r.t inference-time computation and long-horizon task complexity.

The evidence provided includes:
* Theoretical derivation using implicit function theorem.
* Detailed experimental benchmarks with comparative tables (Tables 1-3).
* Clear ablations on feedback types, reuse strategies, and scalability (Figures 5 and 6).

The experiments are clear and the proposed method shows improvement over other approaches.

**Essential References Not Discussed:**

Some of the recent papers that use LLMs for long-horizon planning in both centralized and decentralized fashion, that allow re-planning (and self-verification to some extent):

1. Hongxin Zhang, Weihua Du, Jiaming Shan, Qinhong Zhou, Yilun Du, Joshua B. Tenen- baum, Tianmin Shu, and Chuang Gan. Building cooperative embodied agents modularly with large language models, 2023.
2. Nayak, S., Morrison Orozco, A., Have, M., Zhang, J., Thirumalai, V., Chen, D., ... & Balakrishnan, H. (2024). Long-horizon planning for multi-agent robots in partially observable environments. Advances in Neural Information Processing Systems, 37, 67929-67967.
3. Jun Wang, Guocheng He, and Yiannis Kantaros. Safe Task Planning for Language- Instructed Multi-Robot Systems using Conformal Prediction. arXiv preprint arXiv:2402.15368, 2024.

**Experimental Designs Or Analyses:**

The experiments are well-designed. The authors isolate model contributions via ablation studies (feedback, compute).

However, the evaluation is constrained to a single benchmark (VirtualHome-Env), and further diversity in tasks/environments would have improved generalizability assessment.

**Methods And Evaluation Criteria:**

I actively follow the literature related to robotic task planning and LLM-based reasoning, though I am less familiar with the technical details of deep equilibrium models, which this paper builds upon. From this perspective, the author's core contribution—formulating plan refinement as a fixed-point problem and leveraging DEQ-style training—appears conceptually elegant and well-motivated. However, several aspects of the method raise important questions and concerns when assessed from a robotic planning standpoint.

First, the overall structure of the proposed Equilibrium Sequence Modeling, where planning is treated as a self-refinement process that converges to an equilibrium, fits naturally with how one might think about iterative plan correction in robotics. The idea of conditioning future plan iterations on environmental feedback (closed-loop refinement) is also quite compelling. However, the paper does not offer much intuition about the theoretical behavior of these equilibria—whether they always exist, whether they are unique, and what properties ensure convergence in practical LLM-based settings. These aspects are critical to establishing confidence in the method, especially in robotics, where convergence and stability can directly affect task execution.

Second, while I appreciate the effort to avoid complex reinforcement learning pipelines or externally curated verifiers, I find the training procedure somewhat underexplained from a robotics planning perspective. The authors argue that supervised learning suffices by using the final equilibrium plan $x^*$ as the input and the ground truth plan $y$ as the target, but it is not entirely clear how this helps the model learn to 'refine' in the planning sense, rather than simply mapping one plan to another. In practical terms, what guarantees that the system learns generalizable plan improvement heuristics, rather than overfitting to a specific dataset structure (e.g., VirtualHome-Env)? Also, the method seems to lack explicit regularization mechanisms or consistency constraints across refinement iterations.

Finally, from a planning systems perspective, I would have appreciated more clarity on how well this equilibrium-based method performs when feedback is noisy, delayed, or partially incorrect (scenarios that are quite common in real-world robotic environments). While the use of a world model to estimate internal feedback is a reasonable workaround, the model's robustness to imperfect feedback (or feedback that disagrees with reality) remains unclear.

Overall I like this new perspective in the LLM-based planning space but would appreciate more clarification on these questions.

**Other Comments Or Suggestions:**

* Suggest clarifying the distinction between "plan convergence" and "task success"—sometimes conflated in figures.
* Consider adding visual explanations of failure cases across baselines (Figure 4 could be expanded).

## Update after rebuttal
I wasn't sure if the "official comments" were visible to the authors and hence am including them here:

I appreciate the authors' rebuttal and addressing my concerns in the review. I am assuming that the authors will include some of these changes in the camera ready version (if accepted). I would like to keep my scores the same.

**Other Strengths And Weaknesses:**

Strengths:
1. The paper is conceptually elegant: combining DEQ with planning.
2. Removes reliance on RL while maintaining flexibility and robustness.
3. Strong empirical evidence and reproducibility through open code (did not verify the reproducibility of the code by running it).

Weaknesses:
refer to methods and evaluation section

**Questions For Authors:**

1. How sensitive is performance to the choice of initialization for fixed-point iteration? Would a poor initialization degrade convergence/stability?
2. Have you considered curriculum training—starting from short tasks and scaling up plan length? Would it improve convergence and data efficiency?
3. Can the model generalize to tasks beyond the VirtualHome-Env benchmark without retraining? For instance, can you reuse the same equilibrium modeling process in the ALFRED or RoboTHOR environment (were the prompts tailored heavily for the VirtualHome environment)?
4. Out of curiosity, what are the limitations of Jacobian-free approximation in your experience? Have you quantified the impact of this approximation empirically?
5. Do you anticipate your approach being practical on real-world robot hardware where interaction latencies and imperfect perception are constraints?
Refer to more broader questions in the Methods and Evaluation criteria.

**Relation To Broader Scientific Literature:**

This paper contributes meaningfully at the intersection of LLM-based planning and implicit model training. It extends prior self-refinement work (Welleck et al., 2023; Madaan et al., 2023) by providing a principled optimization framework rather than heuristic prompting. It also offers a scalable alternative to methods like Tree-of-Thoughts (Yao et al., 2023) and SELF-REFINE.

**Theoretical Claims:**

Yes, theoretical claims regarding differentiability through the fixed-point (via the implicit function theorem) are correct and consistent with standard literature (Bai et al., 2019; Krantz & Parks, 2002).

I am not familiar with the Jacobian-free approximation (Fung et al., 2022) though and would urge the Area Chair to rely on the reviews of other reviewers on the correctness of the claim (Line 73).

---

> ### Author Rebuttal · Authors · 2025-03-31
>
> We sincerely appreciate the reviewer for providing valuable feedback. Below are our responses to the raised concerns.
>
> [W1] Existence and uniqueness of equilibrium points
> * We empirically confirm that equilibria always exist across different initializations and tasks (see Table 5 in the appendix). The intuition behind this is that LLMs tend to repeat themselves under greedy sampling, leading to easier convergence.
> * For the uniqueness of the equilibria, we run 10 seeds per task over 60 random tasks. As shown in the table below, our planner typically converges to 2-4 different equilibrium solutions for the same task. However, these solutions are highly consistent in that they either all succeed (SR$\rightarrow$100) or all fail (SR$\rightarrow$0), further confirming the stability of convergence.
>   |Task subset|#Equilibrium solutions|Average SR|
>   |-|--|-|
>   |I|2.080|97.33|
>   |II|3.885|5.91|
>
> [W2] Intuition behind learning to refine
> * Training the model to map the equilibrium plan $x^*$ to the ground truth $y$ is essentially mapping a suboptimal solution to a better solution, which involves self-refinement. Meanwhile, $x^*$ provides a good prior so that the training objective does not collapse to direct regression of the ground truth (as in standard supervised learning).
> * The generalization of self-refinement is facilitated by our training objective, which avoids directly regressing to the ground truth and therefore overfits less to the training data. As further evidence, in the response to W4 below, we validate the generalizability of our method by evaluating it on ALFRED without retraining.
> * The consistency during self-refinement (as shown in Figures 10-14) comes implicitly from the LLM's instruction-following capability and its tendency to repeat itself. We note that it is possible to incorporate explicit rules to generate structured output from LLMs, which may provide further performance improvements.
>
> [W3] Robustness to noisy feedback
> * Please see the response to Reviewer Rz3G's W2.
>
> [W4] Generalization to more benchmarks
> * Following your suggestion, we test pre-trained VirtualHome planners on ALFRED with updated prompts. For quick evaluation, we consider two planning metrics: task classification accuracy (across 7 task types) and recall of ground-truth action-object pairs in the predicted plan. As shown in the table below, our planner generalizes significantly better than the supervised trained planner.
>   |VirtualHome -> ALFRED|Task classification acc.|Action-object recall|
>   |-|-|-|
>   |SFT Llama|11%|0.50%|
>   |Ours|**54%**|**27.08%**|
>
> [W5] Missing references
> * Thank you for pointing out the recent papers on long-horizon planning. We will revise the draft to include their discussion.
>
> [W6] Distinction between "plan convergence" and "task success"
> * "Plan convergence" refers to the convergence of planner output, but this does not necessarily indicate "task success", which requires successful execution of the plan. See Figures 13a, 13b and 14a for failures after plan convergence.
>
> [W7] Visual explanations of baseline failure
> * Thank you for your suggestion. We provide more failure cases of the baselines in Figures 10 and 11 in the appendix, where they fail to correct a long plan efficiently. We will add more visualization explanations in the revision.
>
> [Q1] Sensitivity to initialization
> * To validate the performance stability w.r.t. initialization, we evaluate over 10 initial seeds for each of 60 random tasks. As shown in the results below, our method has a low standard deviation in SR and GCR and outperforms the strongest baseline Tree-Planner, confirming its stability. For convergence sensitivity w.r.t. initialization, see Table 5 in the appendix.
>   |Method|SR|GCR|
>   |-|-|-|
>   |Tree-Planner (N=50)|38.33|56.95|
>   |Ours|**43.50** $\pm$ 5.21|**68.88** $\pm$ 5.94|
>
> [Q2] Curriculum training
> * Following your suggestion, we perform curriculum training on tasks of increasing difficulty. Specifically, the model is trained for six iterations: the first four use subsets of task lengths below 5, 10, 15, and 20, and the last two use the full dataset. As shown in the table below, despite the reduced total data size (by 36%), the model achieves higher SR and comparable GCR in Both novel scenarios. This demonstrates the improved data efficiency of curriculum training.
>   |Curriculum training|Data size|Both|novel||Novel|scene||Novel|task|
>   |-|-|-|-|-|-|-|-|-|-|
>   |||SR|GCR||SR|GCR||SR|GCR|
>   |x|11469|51.61|**75.13**||**75.79**|**85.79**||**56.62**|**75.53**|
>   |$\checkmark$|7411|**54.83**|73.73||69.47|82.93||52.99|71.31|
>
> [Q3] Generalization without retraining
> * Please see the response to W4.
>
> [Q4] Limitation of Jacobian-free approximation
> * Please see the reponse to Reviewer Rz3G's W5.
>
> [Q5] Practicality on real-world robots
> * Please see the response to Reviewer nTaB's W1.

---

### Official Review · Reviewer_Rz3G · 2025-03-14

**Overall Recommendation:** 3

**Summary:**

In this paper, a closed-loop long-horizon robot planning method based on equilibrium sequence modeling is proposed. By treating the planning as a self-optimizing fixed-point solution process, the implicit gradient of the deep equilibrium model is leveraged for end-to-end supervised training without the need for additional validators or reinforcement learning. From an analytical point of view, theoretical tools such as the implicit function theorem are used to optimize the training process and avoid complex backpropagation. In addition, the nested equilibrium mechanism is designed to dynamically allocate computing resources to optimize planning efficiency by combining environmental feedback and world model.

**Claims And Evidence:**

Yes. The claims made in the submission are supported by clear and convincing evidence.

**Essential References Not Discussed:**

The paper has provided a comprehensive review of relevant literature.

**Experimental Designs Or Analyses:**

The experimental designs and analyses in the paper are generally sound and well-structured.

**Methods And Evaluation Criteria:**

The proposed methods and evaluation criteria are well-suited for the problem of closed-loop long-horizon robotic planning. The equilibrium sequence modeling approach, nested equilibrium solving, and world model integration address key challenges in this domain. The use of the Virtual Home-Env benchmark and relevant evaluation metrics provides a robust framework for assessing the method's performance.

**Other Comments Or Suggestions:**

No.

**Other Strengths And Weaknesses:**

Strengths:
1. The paper presents a novel approach by combining the concepts of deep equilibrium models with iterative refinement for robotic planning.
2. The application of the proposed method to the Virtual Home-Env benchmark demonstrates its potential for real-world robotic planning tasks.

Weaknesses:
1. In the training stage, it needs to rely on environmental feedback and real data, which may limit its generalization ability in scenarios where there is a lack of sufficient environmental interaction data. For example, in some real-world robotics tasks, obtaining large amounts of high-quality environmental feedback can be very difficult or costly.
2. In practice, environmental feedback may be disturbed by noise or inaccurate information, does this affect the self-refinement ability and final performance of the model?
3. The article mentions that models can have hallucinatory problems in some cases, such as the equilibrium planner and the world model that may generate plans or feedback that do not correspond to reality. Does this hallucinatory problem lead to planning failures or execution errors, especially in complex robotic tasks?
4. The model currently only supports text input and lacks the ability to process visual information, which limits its applicability in real-world robot tasks. In many real-world scenarios, robots need to process both visual and textual information for accurate planning and decision-making.
5. Although the paper theoretically proposes an optimization method based on equilibrium sequence modeling, in the actual implementation, the authors use approximation methods (such as ignoring the inverse Jacobian matrix) to simplify the training process. The authors should explain whether this approximation will cause the model to fail to achieve theoretically optimal performance in some complex tasks.

**Questions For Authors:**

1. In the process of iterative refinement, how to ensure that the final equilibrium point is the global optimal solution, not the local optimal solution? In theory, the equilibrium point (fixed point) satisfies the condition that the Lipschitz coefficient is less than 1, but how to deal with this problem in practical application to ensure that the sequence reaches convergence?

2. How are the environmental feedback and the world model feedback in the article coordinated and selected in practical applications? In some cases, the feedback from the world model may be skewed from the real-world feedback, will this affect the model's ability to self-refine?

**Relation To Broader Scientific Literature:**

The key contributions of the paper are much related to the field of robotic planning.

**Theoretical Claims:**

Yes, the theoretical claims and proofs in the paper are correct.

---

> ### Author Rebuttal · Authors · 2025-04-01
>
> Thank you for the thoughtful and constructive feedback. Below are our responses to the raised concerns.
>
> [W1] Generalization to environment without feedback
> * There are two workarounds in the absense of environmental feedback: (1) Use zero-shot LLM feedback during training. Since LLMs have been shown effective as zero-shot reward models in reinforcement learning [1], they could similarly provide guidance in our training setup. (2) Pre-train in environments with feedback. We validate this by evaluating pre-trained VirtualHome planners on the new ALFRED environment, where our planner generalizes well to a completely new environment (see reply to Reviewer Bgnq's W4 for details).
>   |VirtualHome -> ALFRED|Task classification acc.|Action-object recall|
>   |-|-|-|
>   |SFT Llama|11%|0.50%|
>   |Ours|**54%**|**27.08%**|
>
> [W2] Robustness to noisy feedback
> * To evaluate our model under noisy feedback, we randomly replace some environmental feedback with incorrect feedback during inference. As shown in the results below, our model exhibits stable performance under small amounts of noise ($\le$10%), demonstrating its robustness.
>   |Noise ratio|Both|novel||Novel|scene||Novel|task|
>   |-|-|-|-|-|-|-|-|-|
>   ||SR|GCR||SR|GCR||SR|GCR|
>   |0%|51.61|**75.13**||**75.79**|85.79||**56.62**|**75.53**|
>   |10%|**53.23**|73.43||74.74|**85.84**||53.85|73.09|
>   |20%|50.00|73.10||73.68|83.22||54.49|71.80|
>
> [W3] Hallucination of planner and world model
> * Hallucination is inherent to LLMs and affects both our planner and world model. However, a key strength of our planner is its ability to self-correct based on feedback. As shown in Figures 10-14 in the appendix, the planner may start with an incorrect plan but improves it using feedback from the environment or the world model, leading to significantly better performance than standard LLM planners (see Tables 1 and 2).
> * For the hallucination of the world model, we show in the reply to W2 that our method is robust to noisy feedback. Thus, even if the world model occasionally hallucinates, the feedback it provides still helps to improve performance (see Table 3).
>
> [W4] Lack of visual information
> * We acknowledge this limitation in Appendix D. While the current method is implemented on LLM planners focused on text input, it could be adapted to video-language planners [2] with visual capabilities. We will continue to investigate this line of planning in future work.
>
> [W5] Impact of Jacobian-free approximation
> * Theoretically, due to the complexity of transformers, the Jacobian-free approximation cannot guarantee convergence to the global optimal solution, which may limit performance. Nevertheless, such approximation is a necessary tradeoff for the computational feasibility of large language models, as demonstrated in [3].
> * Empirically, we confirm that the model can sometimes get stuck in local optimal equilibrium solutions that lead to task failure (see the respone to Q1 below). While there might be several reaons, we suspect that this is partly due to the limitations of the Jacobian-free approximation. Improving this approximation could lead to better performance.
>
> [Q1] Global optimality of equilibrium solution
> * To assess the optimality of equilibrium solutions, we evaluate 10 seeds on each of 60 random tasks. The results show that: (1) our planner can converge to 2-4 different equilibrium solutions for the same task, and the solutions are highly consistent in that they either all succeed (SR→100) or all fail (SR→0), (2) it doesn't always converge to global optimal solutions, as there are still failure cases. This could be due to the limitation of Jacobian-free approximation discussed above.
>   |Task subset|#Equilibrium solutions|Average SR|
>   |-|--|-|
>   |I|2.080|97.33|
>   |II|3.885|5.91|
> * In practice, we find that the equilibrium solving process always converges (see Table 5 in the appendix). This is largely due to the tendency of LLMs to repeat themselves under greedy sampling, which facilitates convergence.
>
> [Q2] Use of environmental and world model feedback
> * Our planner alternates between using environmental feedback and world model feedback at each outer-loop iteration, as illustrated in Figure 12c in the appendix. This allows for better performance within limited environmental interactions. We note that in practical applications, their frequencies could be adjusted according to the accuracy of the world model and the interaction budget.
> * Despite the fact that the world model feedback may be skewed, it still provides useful information for self-refinement, resulting in improved performance (see Table 3) and convergence speed (see reply to Reviewer dmhf's W1). This can be explained by the response to W2, where our method is shown to be robust to noisy feedback.
>
> ---
>
> [1] Ma, et al. Eureka: Human-level reward design via coding large language models. ICLR 2024.
>
> [2] Du, et al. Video language planning. ICLR 2024.
>
> [3] Choe, et al. Making scalable meta learning practical. NeurIPS 2023.

---

### Official Review · Reviewer_dmhf · 2025-03-18

**Overall Recommendation:** 3

**Summary:**

This paper introduces "Equilibrium Sequence Modeling," a novel framework for robotic task planning using Large Language Models (LLMs). The authors propose reformulating plan refinement as a fixed-point problem solvable with deep equilibrium models, enabling a supervised training scheme for improving self-refinement. The fine-tuned model is then used in the proposed framework, iteratively refining the plan while incorporating predicted environmental feedback from a fine-tuned LLM world model. The authors demonstrate improved success rate over several baselines on novel tasks from the VirtualHome dataset.

**Claims And Evidence:**

The paper claims that the proposed method has improved planning performance, better scaling w.r.t inference computation, and effectively incorporates closed-loop feedback.

The evidence for these claims is a set of experiments on the VirtualHome-Env benchmark. The claims of improved performance are supported by the presented results, showing improved success rates over baselines. The results show that the method improves how quickly self-refinement converges, and also how much performance improves with increasing test-time computation. This suggests that the proposed method shows promise for effectively using inference-time resources.

To support the claim that the method effectively incorporates closed-loop feedback, the authors have an ablation which shows that including feedback correction improves planning performance.

The claim that the world model significantly reduces computational cost is plausible, but no data or evidence is given for this claim. The results also show a decrease in executability, which needs further explanation.

**Essential References Not Discussed:**

Guiding long-horizon task and motion planning with vision language models, Z Yang, C Garrett, D Fox, T Lozano-Pérez, LP Kaelbling

Predicate Invention from Pixels via Pretrained Vision-Language Models. Ashay Athalye, Nishanth Kumar, Tom Silver, Yichao Liang, Tomás Lozano-Pérez, Leslie Pack Kaelbling

Generalized Planning in PDDL Domains with Pretrained Large Language Models. Tom Silver, Soham Dan, Kavitha Srinivas, Joshua B. Tenenbaum, Leslie Pack Kaelbling, Michael Katz

**Experimental Designs Or Analyses:**

N/A, sufficiently covered above

**Methods And Evaluation Criteria:**

Using deep equilibrium models to improve the self-refinement process seems to be a reasonable approach. In principle the approach could be applied to domains beyond robot task planning.

W.r.t the claim of incorporating closed-loop feedback, one of the primary motivations for closed-loop feedback is to adapt to environmental disturbances. An experiment showing that incorporating closed-loop feedback better adapts to online disturbances would be much more compelling.

Training the model requires generating a dataset of plans by solving the equilibrium model, which is an expensive process. The quality of this original dataset also depends strongly on the quality of the original model. What if the initial self refinement fails to converge/improve and fails to produce an informative dataset? It would be informative to ablate the performance with different base models of different quality.

**Other Comments Or Suggestions:**

N/A

**Other Strengths And Weaknesses:**

The clarity of the paper could be improved:

- A brief description of the task planning problem in section 3 would be useful for setting the paper up.

- Figure 3 is not very clear to me at all. It contains all proposed components, i.e. the memory, world model, and planner. However it seems there should be a distinction between what is occurring at inference-time and what is occurring at training time. For example, from my understanding the memory is not used during inference-time but is used during training. In addition, what is the interaction between the world model and the environment during training in Figure 3? The meaning of the arrows to and from the world model / environment is not clear.

**Questions For Authors:**

1. Why does using a combination of feedback from the world model and environment result in higher performance than the environment only?
2. The results show that the proposed method scores lower on executability than baselines. The reason given is because of illegal overlength inputs. Why is the proposed method more susceptible to this problem than the baselines?
3. Did the authors consider iteratively training and re-collecting a new dataset using the new refinement model? I am curious about the potential performance ceiling.
4. Did the authors come across any situations where the refinement method fails to converge?
5. How does the quality of the base model affect the dataset generation, and what happens if the initial self-refinement fails to converge?

**Relation To Broader Scientific Literature:**

The paper relates to the broader scientific literature on LLM-based robotic planning and deep equilibrium models.. The use of deep equilibrium models for self-refinement is a novel contribution.

**Theoretical Claims:**

The paper does not present formal proofs for theoretical claims. The core theoretical concept is the formulation of plan refinement as a fixed-point problem. While this is a reasonable approach, there is no analysis of the convergence of the fixed-point problem.

---

> ### Author Rebuttal · Authors · 2025-04-01
>
> Thank you for the very constructive comments. Below, we first address the raised questions.
>
> [Q1] Advantage of combined feedback
> * The world model's feedback offers an alternative to environmental feedback when the interaction budget is limited. By combining both feedbacks, the planner can perform additional self-correction rounds within the same budget. This leads to faster convergence (see response to W1 below) and thus improved performance (see Table 3).
>
> [Q2] Lower executability
> * The lower executability of our method is due to format issues in overlength outputs. This is likely caused by complex prompts (including history plans and feedback) or limited memorization of ground truths. However, these errors are minor and can be fixed with simple post-processing, resulting in nearly 100% executable plans in the virtual environment.
>
> [Q3] Iterative training with new dataset
> * This is also how we train the model: after each iteration, we update the training dataset with the equilibrium solutions generated by the newly finetuned planner, and use this updated dataset for the next training iteration. As shown in the table below, the overall performance steadily improves over training iterations.
>   |#Iterations|Both|novel||Novel|scene||Novel|task|
>   |-|-|-|-|-|-|-|-|-|
>   ||SR|GCR||SR|GCR||SR|GCR|
>   |0|00.00|00.00||00.00|00.00||00.00|00.00|
>   |2|51.61|74.67||65.26|82.78||**60.68**|**78.40**|
>   |4|51.61|73.30||68.42|82.93||53.20|72.81|
>   |6|**51.61**|**75.13**||**75.79**|**85.79**||56.62|75.53|
>
> [Q4] Convergence analysis
> * We observe that self-refinement consistently converges across different initializations. The table below summarizes the number of iterations to convergence over 10 runs on 60 random tasks, where all tested LLMs (including two new Qwen models) converge to an equilibrium within a finite number of iterations. The intuition behind this is that LLMs tend to repeat themselves, leading to easier convergence.
>   |Method|Mean|Std|Min|Max|
>   |-|-|-|-|-|
>   |Original Llama-3-8B|6.70|2.12|3.22|14.98|
>   |Original Qwen-2.5-0.5B|4.80|2.13|2.69|9.43|
>   |Original Qwen-2.5-7B|7.68|5.03|2.46|16.78|
>   |SFT Llama-3-8B|2.46|0.88|2.02|3.80|
>   |Ours|3.02|1.61|2.32|4.07|
>
> [Q5] Dataset generation w.r.t. base model
> * Since self-refinement consistently converges in practice, the resulting equilibrium can be reliably paired with ground truth to form meaningful training data. This holds true for both Qwen and Llama base models.
>
> ---
>
> We also summarize other weaknesses mentioned in the review and provide our clarifications below.
>
> [W1] Claim of reducing computational cost
> * To clarify, we did not claim that the world model reduces computational cost. Instead, its goal is to reduce the number of environmental interactions, as mentioned in Section 3.4. To verify this claim, we compare the evolution of the success rate (SR) with and without the world model in the table below, which shows that the world model allows a similar or higher success rate even with fewer environmental interactions.
>   |#Interactions|World model|Both novel|Novel scene|Novel task|
>   |-|-|-|-|-|
>   |0|x|33.87|49.47|34.62|
>   ||$\checkmark$|**37.09**|**65.26**|**40.17**|
>   |1|x|50.00|71.57|**53.41**|
>   ||$\checkmark$|**53.22**|**75.78**|50.85|
>   |2|x|51.61|74.73|**55.12**|
>   ||$\checkmark$|**56.45**|**76.84**|53.63|
>   |3|x|51.61|75.79|**56.62**|
>   ||$\checkmark$|**56.45**|**77.89**|54.91|
>
> [W2] Robustness to environmental disturbances
> * We simulate environmental disturbances by randomly replacing some environmental feedback with wrong feedback during inference. As shown in the results below, our model exhibits stable performance under small amounts of disturbances ($\le$10%), demonstrating its robustness.
>   |Disturbances|Both|novel||Novel|scene||Novel|task|
>   |-|-|-|-|-|-|-|-|-|
>   ||SR|GCR||SR|GCR||SR|GCR|
>   |0%|51.61|**75.13**||**75.79**|85.79||**56.62**|**75.53**|
>   |10%|**53.23**|73.43||74.74|**85.84**||53.85|73.09|
>   |20%|50.00|73.10||73.68|83.22||54.49|71.80|
>
> [W3] Missing references
> * Thank you for pointing out the related work on planning with VLMs and LLMs. We will update the draft to include their discussion.
>
> [W4] Description of task planning problem
> * Thanks for the valuable suggestion. We will add a brief description of the task planning problem at the beginning of Section 3 to improve clarity.
>
> [W5] Clarity of Figure 3
> * Figure 3 illustrates our planning framework during training, and thus includes training-specific components, e.g. the equilibrium memory you noted. The world model does not directly interact with the environment; instead, it is trained on past experiences stored in the equilibrium memory, as shown by the arrow pointing to it. The single arrow from the environment (and none from the world model) indicates that the planner receives feedback solely from the environment during training. We will revise the figure description to make this clearer.

---

### Official Review · Reviewer_nTaB · 2025-03-19

**Overall Recommendation:** 3

**Summary:**

This paper proposes an  equilibrium model-based planner for decomposing high-level tasks into mid-level action sequences in an iterative manner taking environment and world model feedback. Experiments on VirtualHome-Env benchmark demonstrates that the approach can improve over a few existing approaches.

**Claims And Evidence:**

Yes.

**Essential References Not Discussed:**

No, as I am aware of.

**Experimental Designs Or Analyses:**

Yes.

**Methods And Evaluation Criteria:**

Yes, but it would be better if  the proposed approach could be grounded from high level tasks and mid-level actions on to level real robot executable primitive actions or even control.

**Other Comments Or Suggestions:**

1. Figure 3, it would be better to clarify whether it is an inference framework, training framework or both. As it is now, the input-output relationship of the planner is not clear: no output. And how the output of the planner gets to the environment and world model, how the planner get feedback from them should be depict clearly.  Similarly,  for section 3.4 practical implementation, it would be better to specify the training and inference pipelines separately for clarification reason.
	2. In the description for table 3, please clarify or define how the world model and environment feedbacks enter the model during training and inference time separately for each combination.

**Other Strengths And Weaknesses:**

Strength:
	1. Introducing equilibrium sequence modeling for planning with LLMs.
	2. With iterative refinement planning paradigm, environment feedbacks and world model feedbacks can be taken into the planning process.

Weakness:
	1. Haven't connect the approach to real-world robot control yet to fully verify the validity of the proposal.
	2. Successful rate is still low towards real application.

**Questions For Authors:**

As mentioned above.

**Relation To Broader Scientific Literature:**

Planning with LLMs or multi-modality foundation models is an important topic to push the AI frontier with physical interaction. This work depicts one way to do so in an iteratively refinement manner and improve over a few existing planning algorithms/paradigms using LLMs as backbone.

**Theoretical Claims:**

Yes, the proofs in the appendix  from the literature.

---

> ### Author Rebuttal · Authors · 2025-04-01
>
> We deeply appreciate your valuable suggestions, and we would like to address your main concerns as follows:
>
> [W1] Connection to real-world robot control
>
> * Our work focuses on high-level planning that decomposes each task into mid-level actions, as a complementary direction to low-level control. Extending this to real-world robotics with low-level actions presents several key challenges: (1) effective tokenization of actions and visual inputs, requiring progress in vision-language action models, (2) handling latency from the equilibrium solving process. To address the latter, our method includes designs such as reusing equilibrium solutions to reduce overhead.
>
> [W2] Low success rate towards application
>
> * The lower success rate can be attributed to three main factors: (1) limited data in the VirtualHome-Env dataset, with only 1360 action sequence annotations, (2) relatively small model size (Llama 3 8B), which may constrain capability, (3) the absence of multimodal inputs that humans naturally rely on. We expect significant performance gains by increasing model size and multimodal data.
>
> [W3] Clarity of Figure 3 and Section 3.4
>
> * Figure 3 illustrates our training framework. At each step, the planner takes context $c_t$ as input and outputs a plan $x_t$. Only the subset of equilibrium plans interact with the environment and are stored in the equilibrium memory. During training, the planner receives feedback solely from the environment based on these equilibrium plans.
>
> * The training and inference pipelines are detailed separately in Appendix B.3. Both the planner and world model are trained using environmental feedback stored in the equilibrium memory. They then work together during inference, where the world model provides additional feedback to the planner.
>
> [W4] Use of world model and environmental feedback in Table 3
>
> * During training, we use only environmental feedback to train both the planner and the world model. At inference time, we evaluate the following four setups:
>   1. No feedback: Only inner-loop refinements without external feedback.
>   2. Environmental feedback: Up to 10 rounds, provided after each inner loop converges.
>   3. World model feedback: also up to 10 rounds, provided after inner loop convergence.
>   4. Both feedback: Alternating between environmental and world model feedback after each inner loop, within a total budget of 10 environmental interactions.

---

### Decision · Program_Chairs · 2025-05-01

**Decision:**

Accept (poster)

**Comment:**

This paper introduces a novel framework, Equilibrium Sequence Modeling, applying deep equilibrium models to LLM-based robotic task planning for iterative self-refinement. The reviews are generally positive (ranging from weak accept to accept), recognizing the conceptual novelty and the empirical evidence demonstrating improved performance over baselines on the VirtualHome-Env benchmark.

Reviewers highlighted the strengths of the principled approach to supervised training for refinement without RL and the effective integration of feedback. The authors provided a comprehensive rebuttal. They included additional experiments addressing robustness, generalization to a new environment (ALFRED), convergence analysis, and the benefits of curriculum training, along with clarifications on the methodology and results. These responses appear to have addressed the major points raised.

Given the novel contribution, the supportive experimental results, and the thorough author response addressing reviewer concerns, the paper is recommended for acceptance.